# Extrapolation Towards Imaginary $0$-Nearest Neighbour and Its Improved Convergence Rate

Akifumi Okuno[1,3] and Hidetoshi Shimodaira[2,3]

[1]School of Statistical Thinking, The Institute of Statistical Mathematics
[2]Graduate School of Informatics, Kyoto University
[3]RIKEN Center for Advanced Intelligence Project

## Abstract

$k$-nearest neighbour ($k$-NN) is one of the simplest and most widely-used methods for supervised classification, that predicts a query's label by taking weighted ratio of observed labels of $k$ objects nearest to the query. The weights and the parameter $k \in \mathbb{N}$ regulate its bias-variance trade-off, and the trade-off implicitly affects the convergence rate of the excess risk for the $k$-NN classifier; several existing studies considered selecting optimal $k$ and weights to obtain faster convergence rate. Whereas $k$-NN with non-negative weights has been developed widely, it was also proved that negative weights are essential for eradicating the bias terms and attaining optimal convergence rate. In this paper, we propose a novel *multiscale $k$-NN (MS-$k$-NN)*, that extrapolates unweighted $k$-NN estimators from several $k \geq 1$ values to $k = 0$, thus giving an imaginary 0-NN estimator. Our method implicitly computes optimal real-valued weights that are adaptive to the query and its neighbour points. We theoretically prove that the MS-$k$-NN attains the improved rate, which coincides with the existing optimal rate under some conditions.

## 1 Introduction

Supervised classification has been a fundamental problem in machine learning and statistics over the years. It is widely used in a number of applications, such as music-genre categorization (Li et al., 2003), medical diagnosis (Soni et al., 2011), speaker recognition (Ge et al., 2017) and so forth. Moreover, vast amounts of data have become readily available for anyone, along with the development of information technology; potential demands for better classification are still growing.

One of the simplest and most widely-used methods for supervised classification is $k$-nearest neighbour ($k$-NN; Fix & Hodges (1951)), where the *estimator* predicts a query's label probability by taking the weighted ratio of observed labels of $k$ objects nearest to the query, and the corresponding *classifier* specifies the class of objects via the predicted label probabilities. $k$-NN has strengths in its simplicity and flexibility over and above its statistical consistency (as $k = k_n \to \infty, k_n/n \to 0, n \to \infty$), proved by Fix & Hodges (1951), Cover & Hart (1967) and Stone (1977). However, such a simple

Table 1: Convergence Rates

|  |  |
|---|---|
| Nadaraya-Watson | $n^{-4/(4+d)}$ |
| Local polynomial[†] | $n^{-2\beta/(2\beta+d)}$ |
| $k$-NN (unweighted) | $n^{-4/(4+d)}$ |
| $k$-NN (with weights $\geq 0$) | $n^{-4/(4+d)}$ |
| $k$-NN (with weights $\in \mathbb{R}$) | $n^{-2\beta/(2\beta+d)}$ |
| **Multiscale $k$-NN** | $n^{-2\beta/(2\beta+d)}$ |

[†]uniform bound; others are non-uniform.
$\alpha = 1, \beta = 2u, u \in \mathbb{N}, \gamma = 2.$

$k$-NN heavily depends on the selection of parameters, i.e., the weights and $k$ therein; inexhaustible discussions on parameter selection have been developed for long decades (Devroye et al., 1996; Boucheron et al., 2005; Audibert & Tsybakov, 2007; Samworth, 2012; Chaudhuri & Dasgupta, 2014; Anava & Levy, 2016; Cannings et al., 2017; Balsubramani et al., 2019).

A prevailing line of research in the parameter selection focuses on *misclassification error rate* of classifiers as the sample size $n$ grows asymptotically. They attempt to minimize the *convergence rate* of the *excess risk*, i.e., the difference of error rates between the classifier and Bayes-optimal classifier . The convergence rate depends on the functional form of the conditional expectation $\eta(x) := \mathbb{E}(Y \mid X = x)$ of the binary label $Y \in \{0, 1\}$ given its feature vector $X \in \mathcal{X} (\subset \mathbb{R}^d)$. Its function class is specified by (i) $\alpha$-**margin condition**, (ii) $\beta$-**Hölder condition**, (iii) $\gamma$-**neighbour average smoothness**, that will be formally described in Definition 1, 2 and 3 later in Section 2. Roughly speaking, classification problems with larger $\alpha \geq 0, \beta > 0, \gamma > 0$ values are easier to solve, and the corresponding convergence rate becomes faster; the rates for specific cases are summarized in Table 1.

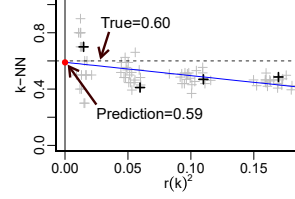

Figure 1: For a fixed query $X_* \in \mathbb{R}^5$, unweighted $k$-NN estimators using synthetic data for 4 different $k$ values are plotted as grey points (20 times); bias-variance trade-off is observed. In an instance shown as black points, $k$-NN estimators are extrapolated to imaginary 0-NN by regression (11), via $r(k)^2 := \|X_{(k)} - X_*\|_2^2$ where $X_{(k)}$ is the $k$-th nearest to the query $X_*$.

For unweighted $k$-NN, Chaudhuri & Dasgupta (2014) proves the rate $O(n^{-(1+\alpha)\gamma/(2\gamma+d)})$ by imposing $\alpha$-margin condition and $\gamma$-average neighbour smoothness. Whereas the rate seems favorable, $\gamma$ is upper-bounded by 2 due to the asymptotic bias, even if highly-smooth function is considered ($\beta \geq 2$; our Theorem 1). Thus the rate for unweighted $k$-NN is $O(n^{-2(1+\alpha)/(4+d)})$ at best.

$k$-NN estimator has much in common with Nadaraya-Watson (NW) estimator (Tsybakov, 2009), and its classifier is proved to attain the same rate $O(n^{-4/(4+d)})$ as unweighted $k$-NN, for $\alpha = 1$ and twice-differentiable $\eta$ (Hall & Kang, 2005). It is also widely known that the convergence rate of local polynomial (LP)-estimator (Tsybakov, 2009) is drastically improved from that of the NW-estimator, when approximating highly smooth functions; Audibert & Tsybakov (2007) considers a classifier based on LP-estimator and an *uniform bound* of the excess risk over all the possible $\eta$ and the distribution of $X$. The rate for LP classifier is $O(n^{-(1+\alpha)\beta/(2\beta+d)})$, which is also proved to be optimal among all the classifiers. However, the LP estimator employs polynomials of degree $\lfloor \beta \rfloor := \max\{\beta' \in \mathbb{N}_0 \mid \beta' < \beta\}$; it estimates coefficients of $1 + d + d^2 + \cdots + d^{\lfloor \beta \rfloor}$ terms, resulting in high computational cost and difficulty in implementation.

Returning back to $k$-NN classifiers, which do not require such a large number of coefficients therein, Samworth (2012) finds optimal weights for weighted $k$-NN by minimizing the exact asymptotic expansion of the non-uniform bound of the excess risk. When considering only non-negative weights, optimal convergence rate is $O(n^{-4/(4+d)})$, where the rate is still same as the case $\alpha = 1$ of the unweighted $k$-NN. However, interestingly, Samworth (2012) also proves that **real-valued weights** including negative weights are essential for eradicating the bias and attaining the exact optimal rate $O(n^{-2\beta/(2\beta+d)})$ for $\eta \in C^\beta$ with $\alpha = 1, \beta = 2u \ (u \in \mathbb{N})$.

**Current issue:** In practice, determining the weights explicitly in the way of Samworth (2012) is rather burdensome, where explicit weights are shown for limited cases ($\beta = 2, 4$). Other simpler approaches to determine optimal weights are appreciated.

**Contribution of this paper:** We propose *multiscale k-NN (MS-k-NN)*, consisting of two simple steps: (1) unweighted $k$-NN estimators are computed for several $k \geq 1$ values, and (2) extrapolating them to $k = 0$ via regression, as explained in Figure 1. This algorithm eradicates the asymptotic bias, as it computes an imaginary 0-NN estimator. Whereas the MS-$k$-NN is computed quite simply, it corresponds to the weighted $k$-NN equipped with favorable real-valued weights, which are automatically specified via regression. Our algorithm implicitly computes the optimal weights adaptive to the query and its neighbour points. We prove that the MS-$k$-NN attains the improved convergence rate $O(n^{-(1+\alpha)\beta/(2\beta+d)})$, that coincides with the optimal rate obtained in Samworth (2012) if $\alpha = 1, \beta = 2u \ (u \in \mathbb{N})$. Numerical experiments are conducted for performing MS-$k$-NN.

We last note that, the weights implicitly obtained in MS-$k$-NN are different from those of Samworth (2012), though both of these weights attain the same optimal convergence rate. See Figure 2 in Section 4.2 for the obtained weights. Also see Supplement A for remaining related works.

## 2 Preliminaries

We describe the problem setting, notation, and the conditions in Section 2.1, 2.2, 2.3, respectively.

### 2.1 Problem setting

For a non-empty compact set $\mathcal{X} \subset \mathbb{R}^d$, $d \in \mathbb{N}$, a pair of random variables $(X, Y)$ takes values in $\mathcal{X} \times \{0, 1\}$ with joint distribution $\mathbb{Q}$, where $X$ represents a feature vector of an object, and $Y$ represents its binary class label to which the object belongs. $\mu$ represents the probability density function of $X$ and $\eta$ is the conditional expectation

$$\eta(x) = \mathbb{E}(Y \mid X = x). \tag{1}$$

Conditions on $\eta$ are listed in Definition 1–3 later in Section 2.3.

$\mathcal{D}_n := \{(X_i, Y_i)\}_{i=1}^n$, $n \in \mathbb{N}$, and $(X_*, Y_*)$ are considered throughout this paper, where they are independent copies of $(X, Y)$; $\mathcal{D}_n$ is called a *sample*, and $X_*$ is called a *query*. Given a query $X_* \in \mathcal{X}$, we consider predicting the corresponding label $Y_*$ by a *classifier* $\hat{g}_n : \mathcal{X} \to \{0, 1\}$ using the sample $\mathcal{D}_n$. The performance of a classifier $g$ is evaluated by the misclassification error rate $L(g) := \mathbb{P}_{X_*, Y_*}(g(X_*) \neq Y_*)$. Under some mild assumptions, *excess risk*

$$\mathcal{E}(\hat{g}_n) := \mathbb{E}_{\mathcal{D}_n}(L(\hat{g}_n)) - \inf_{g:\mathcal{X} \to \{0,1\}} L(g) \tag{2}$$

for various classifiers is proved to approach $0$ as $n \to \infty$. Note that the classifier $g_*(X) := \mathbb{1}(\eta(X) \geq 1/2)$ satisfies $L(g_*) = \inf_{g:\mathcal{X} \to \{0,1\}} L(g)$, and it is said to be Bayes-optimal (see, e.g., Devroye et al., 1996, Section 2.2). Then, the asymptotic order of the excess risk $\mathcal{E}(\hat{g}_n)$ with respect to the sample size $n$ is called *convergence rate*; the goal of this study is to propose a classifier that (i) is practically easy to implement, and (ii) attains the optimal convergence rate.

### 2.2 Notation

For any given query $X_* \in \mathcal{X}(\subset \mathbb{R}^d)$ and a sample $\mathcal{D}_n$, the index $1, 2, \ldots, n$ is re-arranged to be $(1), (2), \ldots, (n)$ s.t. $\|X_* - X_{(1)}\|_2 \leq \|X_* - X_{(2)}\|_2 \leq \cdots \leq \|X_* - X_{(n)}\|_2$ where Euclidean norm $\|\boldsymbol{x}\|_2 = (\sum_{i=1}^d x_i^2)^{1/2}$ is employed throughout this paper. Note that the re-arranged index $(1), (2), \ldots, (n)$ depends on the query $X_*$; we may also denote the index by $(1; X_*), (2; X_*), \ldots, (n; X_*)$. $B(X; r) := \{X' \in \mathcal{X} \mid \|X - X'\|_2 \leq r\} \subset \mathcal{X}$ represents the $d$-dimensional closed ball centered at $\boldsymbol{x} \in \mathcal{X}$ whose radius is $r > 0$.

$f(n) \asymp g(n)$ indicates that the asymptotic order of $f, g$ are the same, $\text{tr}\boldsymbol{A} = \sum_{i=1}^d a_{ii}$ represents the trace of the matrix $\boldsymbol{A} = (a_{ij}) \in \mathbb{R}^{d \times d}$, $\mathbf{1} = (1, 1, \ldots, 1)^\top$ is a vector and $\mathbb{1}(\cdot)$ represents an indicator function. $\lfloor \beta \rfloor := \max\{\beta' \in \mathbb{N} \mid \beta' < \beta\}$ for $\beta > 0$, $[n] := \{1, 2, \ldots, n\}$ for any $n \in \mathbb{N}$, and $\|\boldsymbol{x}\|_\infty := \max_{i \in [d]} |x_i|$ for $\boldsymbol{x} = (x_1, x_2, \ldots, x_d)$. Let $\mathbb{N}_0 = \mathbb{N} \cup \{0\}$. For $q \in \mathbb{N}_0$, $C^q = C^q(\mathcal{X})$ represents a set of $q$-times continuously differentiable functions $f : \mathcal{X} \to \mathbb{R}$.

### 2.3 Conditions

We first list three different types of conditions on the conditional expectation (1), in Definition 1, 2 and 3 below; they are considered in a variety of existing studies (Györfi, 1981; Devroye et al., 1996; Audibert & Tsybakov, 2007; Tsybakov, 2009; Chaudhuri & Dasgupta, 2014).

**Definition 1** ($\alpha$-margin condition). If there exist constants $L_\alpha \geq 0, \tilde{t} > 0$ and $\alpha \geq 0$ such that

$$\mathbb{P}(|\eta(X) - 1/2| \leq t) \leq L_\alpha t^\alpha,$$

for all $t \in (0, \tilde{t}]$ and $X \in \mathcal{X}$, $\eta$ is said to be satisfying $\alpha$-*margin condition*, with *margin exponent* $\alpha$.

**Definition 2** ($\beta$-Hölder condition). Let $\mathcal{T}_{q, X_*}[\eta]$ be the Taylor expansion of a function $\eta$ of degree $q \in \mathbb{N}_0$ at $X_* \in \mathcal{X}$ (See, Definition 8 in Supplement for details). A function $\eta \in C^{\lfloor \beta \rfloor}(\mathcal{X})$ is said to be $\beta$-*Hölder*, where $\beta > 0$ is called *Hölder exponent*, if there exists $L_\beta > 0$ such that

$$|\eta(X) - \mathcal{T}_{\lfloor \beta \rfloor, X_*}[\eta](X)| \leq L_\beta \|X - X_*\|^\beta \tag{3}$$

for any $X, X_* \in \mathcal{X}$. Note that a function $\eta \in C(\mathcal{X})^\beta$ for $\beta \in \mathbb{N}$ and compact $\mathcal{X}$ is also $\beta$-Hölder.

The above Hölder condition specifies the smoothness of $\eta$ by a user-specified parameter $\beta > 0$, and the above (3) is employed in many studies, e.g., Audibert & Tsybakov (2007). It reduces to

$$|\eta(X) - \eta(X_*)| \leq L_\beta \|X - X_*\|^\beta \tag{4}$$

for $0 < \beta \leq 1$. However, (3) and (4) are different for $\beta > 1$, where the latter is considered in Chaudhuri & Dasgupta (2014).

For describing the next condition, we consider $\eta^{(\infty)}(B) := \mathbb{E}(Y|X \in B)$, that is the conditional expectation of $Y$ given $X \in B$ for the set $B \subset \mathbb{R}^d$. $\eta^{(\infty)}$ and the support of $\mu$ are expressed as

$$\eta^{(\infty)}(B) = \frac{\int_{B \cap \mathcal{X}} \eta(X)\mu(X)\mathrm{d}X}{\int_{B \cap \mathcal{X}} \mu(X)\mathrm{d}X}, \quad \mathcal{S}(\mu) := \left\{ X \in \mathcal{X} \mid \int_{B(X;r)} \mu(X)\mathrm{d}X > 0, \forall r > 0 \right\} \tag{5}$$

where Chaudhuri & Dasgupta (2014) Lemma 9 proves that $\eta^{(\infty)}(B(X_*;r))$ for $X_* \in S(\mu)$ asymptotically approximates the unweighted $k$-NN estimator (with roughly $r = \|X_{(k)} - X_*\|_2$), that will be formally defined in Definition 5. $\mathcal{S}(\mu)$ is assumed to be compact throughout this paper.

**Definition 3** ($\gamma$-neighbour average smoothness)**.** If there exists $L_\gamma, \gamma > 0$ such that

$$|\eta^{(\infty)}(B(X;r)) - \eta(X)| \leq L_\gamma r^\gamma$$

for all $r > 0$ and $X \in \mathcal{S}(\mu)$, then the function $\eta$ is said to be $\gamma$-*neighbour average smooth* with respect to $\mu$, where $\gamma$ is called *neighbour average exponent*. A weaker version of this condition is used in Györfi (1981), where the constant $L_\gamma$ is replaced by a function $L_\gamma(X)$. A related but different condition called "$(\alpha, L)$-smooth" is used in Chaudhuri & Dasgupta (2014); see Supplement C.

We last define an assumption on the density of $X$, that is employed in Audibert & Tsybakov (2007).

**Definition 4** (Strong density assumption)**.** If there exist $\mu_{\min}, \mu_{\max} \in (0, \infty)$ such that $\mu_{\min} \leq \mu(X) \leq \mu_{\max}$ for all $X \in \mathcal{X}$, $\mu$ is said to be satisfying *strong density assumption*.

## 3  $k$-NN classifier and convergence rates

In Section 3.1, we first define $k$-NN classifier. Subsequently, we review existing studies on convergence rates for unweighted $k$-NN and weighted $k$-NN classifiers in Section 3.2 and 3.3, respectively. Other classifiers and their convergence rates are also presented in Supplement B.

### 3.1  $k$-NN classifier

In this paper, we consider only a plug-in classifier (Audibert & Tsybakov, 2007)

$$g^{(\text{plug-in})}(X; \hat{\eta}_n) := \mathbb{1}(\hat{\eta}_n(X) \geq 1/2), \tag{6}$$

where $\hat{\eta}_n(X)$ is an estimator of $\eta(X)$, that leverages the sample $\mathcal{D}_n$. Given a query $X_* \in \mathcal{X}$, an archetypal example of the function value $\hat{\eta}_n(X_*)$ is in the following.

**Definition 5.** Weighted $k$-NN estimator is defined as

$$\hat{\eta}_{n,k,\boldsymbol{w}}^{(k\text{NN})}(X_*) := \sum_{i=1}^{k} w_i Y_{(i;X_*)}, \tag{7}$$

where $(1; X_*), (2; X_*), \ldots, (n; X_*)$ is the re-arranged index defined in Section 2.2 and $k \in \mathbb{N}$ is a user-specified parameter. It is especially called unweighted $k$-NN if $w_1 = w_2 = \cdots = w_k = 1/k$, and is denoted by $\hat{\eta}_{n,k}^{(k\text{NN})}$. The (weighted) $k$-NN classifier is $\hat{g}_{n,k,\boldsymbol{w}}^{(k\text{NN})}(X) := g^{(\text{plug-in})}(X; \hat{\eta}_{n,k,\boldsymbol{w}}^{(k\text{NN})})$.

### 3.2  Convergence rate for unweighted $k$-NN classifier

Here, we consider the unweighted $k$-NN; the following Proposition 1 shows the convergence rate.

**Proposition 1** (A slight modification of Chaudhuri & Dasgupta (2014) Th. 4)**.** Let $\mathcal{X}$ be a compact set, and assuming that (i) $\eta$ satisfies $\alpha$-margin condition and is $\gamma$-neighbour average smooth, and (ii) $\mu$ satisfies strong density assumption. Then, the convergence rate of the unweighted $k$-NN classifier with $k_* = k_n \asymp n^{2\gamma/(2\gamma+1)}$ is

$$\mathcal{E}(\hat{g}_{n,k_*}^{(k\text{NN})}) = O(n^{-(1+\alpha)\gamma/(2\gamma+d)}).$$

*Proof.* Chaudhuri & Dasgupta (2014) Theorem 4(b) shows the convergence rate; see Supplement C for the correspondence of the assumption and symbols. □

Our current concern is whether the convergence rate $O(n^{-(1+\alpha)\gamma/(2\gamma+d)})$ of the unweighted $k$-NN classifier can be associated to the rate $O(n^{-(1+\alpha)\beta/(2\beta+d)})$ of the LP classifier, whose optimality is proved by Audibert & Tsybakov (2007) and is formally described in Proposition 3 in Supplement B. Chaudhuri & Dasgupta (2014) asserts that these rates are the same, i.e., $\gamma = \beta$, if there exists $L_\beta > 0$ such that (4) holds for any $X, X_* \in \mathcal{X}$. However, only constant functions can satisfy the condition (4) for $\beta > 1$ (Mittmann & Steinwart, 2003, Lemma 2.3); only an extremely restricted function class is considered in Chaudhuri & Dasgupta (2014).

We here return back to the $\beta$-Hölder condition (3) considered in this paper and Audibert & Tsybakov (2007), that is compatible with the condition (4) for $\beta \leq 1$ but is different for $\beta > 1$. Whereas a variety of functions besides constant functions satisfy the $\beta$-Hölder condition (3), our following Theorem 1 shows that $\gamma = 2$ even if $\eta$ is highly smooth ($\beta \gg 2$). Especially for $\alpha = 1$, the rate of unweighted $k$-NN coincides with the rate $O(n^{-4/(4+d)})$ of NW-classifier (Hall & Kang, 2005).

**Theorem 1.** Let $\mathcal{X}$ be a compact set, and let $\beta > 0$. Assuming that (i) $\mu$ and $\eta\mu$ are $\beta$-Hölder, and (ii) $\mu$ satisfies the strong density assumption, there exist constants $L_\beta^* > 0, \tilde{r} > 0$ and continuous functions $b_1^*, b_2^*, \ldots, b_{\lfloor \beta/2 \rfloor}^*, \delta_{\beta,r} : \mathcal{X} \to \mathbb{R}$ such that

$$\eta^{(\infty)}(B(X_*; r)) - \eta(X_*) = \sum_{c=1}^{\lfloor \beta/2 \rfloor} b_c^*(X_*) r^{2c} + \delta_{\beta,r}(X_*), \quad |\delta_{\beta,r}(X_*)| \leq L_\beta^* r^\beta$$

for all $r \in (0, \tilde{r}], X_* \in \mathcal{S}(\mu)$ defined in (5). For $\beta > 2$, $b_1^*(X_*) = \frac{1}{2d+4} \frac{1}{\mu(X_*)} \{\Delta[\eta(X_*)\mu(X_*)] - \eta(X_*)\Delta\mu(X_*)\}$ with $\Delta := \frac{\partial^2}{\partial x_1^2} + \cdots + \frac{\partial^2}{\partial x_d^2}$; if $\eta$ is $\beta(\geq 2)$-Hölder, $\eta$ is $(\gamma =)2$-neighbour average smooth and

$$\mathcal{E}(\hat{g}_{n,k_*}^{(k\mathrm{NN})}) = O(n^{-2(1+\alpha)/(4+d)}).$$

*Proof.* The numerator and denominator of $\eta^{(\infty)}(B(X_*; r))$ are obtained via integrating Taylor expansions of $\eta\mu$ and $\mu$, respectively; division proves the assertion. See, Supplement F for details. □

### 3.3 Convergence rate for weighted $k$-NN classifier

Here, we consider the weighted $k$-NN. Samworth (2012) first derives non-negative optimal weights

$$w_i^* := \frac{1}{k_*} \left\{ 1 + \frac{d}{2} - \frac{d}{2k_*^{2/d}}(i^{1+2/d} - (i-1)^{1+2/d}) \right\} \tag{8}$$

for $i \in [k_*]$ and 0 otherwise, where $k_* \asymp n^{4/(d+4)}$, through the asymptotic expansion of the excess risk. However, the obtained rate is still $O(n^{-4/(4+d)})$ (Samworth (2012) Theorem 2), that is the same as the case $\alpha = 1$ of unweighted $k$-NN (Theorem 1); convergence evaluation of the $k$-NN still remains slow, even if arbitrary weights can be specified.

For further improving the convergence rate, Samworth (2012) also considers **real-valued weights** allowing negative values. The improved convergence rate is given in the following Proposition 2. Formal descriptions, i.e., definition of the weight set $\mathcal{W}_{n,s}$ and conditions for their rigorous proof, are described in Supplement D due to the space limitation.

**Proposition 2** (Samworth (2012) Th. 6)**.** Let $\mathcal{W}_{n,s}$ be a set of real-valued weights defined in Supplement D, where we assume the conditions (i)–(iv) therein. Note that the condition (ii) implies $\eta \in C^\beta, \beta = 2u$ for $u \in \mathbb{N}$. If $(\alpha =)1$-margin condition is assumed, then

$$\mathcal{E}(\hat{g}_{n,k,\boldsymbol{w}}^{(k\mathrm{NN})}) \asymp \left\{ B_1 \sum_{i=1}^n w_i^2 + B_2 \left( \sum_{i=1}^n \frac{\delta_i^{(u)} w_i}{n^{2r/d}} \right)^2 \right\} (1 + o(1)) \tag{9}$$

holds for $\boldsymbol{w} \in \mathcal{W}_{n,s}$, where $B_1, B_2$ are some constants and $\delta_i^{(\ell)} := i^{1+2\ell/d} - (i-1)^{1+2\ell/d} (\forall \ell \in [u])$.

Following Proposition 2, Samworth (2012) shows that the asymptotic minimizer of the excess risk (9) with the weight constraint $\sum_{i=1}^{n} w_i = 1$, $\sum_{i=1}^{n} \delta_i^{(\ell)} w_i = 0$ for all $\ell \in [u-1]$, and $w_i = 0$ for $i = k^* + 1, \ldots, n$ with $k^* \asymp n^{2\beta/(2\beta+d)}$ is in the form of

$$w_i^* := (a_0 + a_1 \delta_i^{(1)} + \cdots + a_u \delta_i^{(u)})/k_* \tag{10}$$

for $i = 1, 2, \ldots, k_*$, where $\boldsymbol{a} = (a_0, a_1, \ldots, a_u) \in \mathbb{R}^{u+1}$ are unknowns. Samworth (2012) proposes to find $\boldsymbol{a}$ so that (10) satisfies the weight constraint. Then, the following optimal rate is obtained.

**Corollary 1.** Symbols and assumptions are the same as those of Proposition 2. Then, the optimal $\boldsymbol{w}_*$ and $k_* \asymp n^{2\beta/(2\beta+d)}$ lead to

$$\mathcal{E}(\hat{g}_{n,k_*,\boldsymbol{w}_*}^{(k\mathrm{NN})}) \asymp n^{-2\beta/(2\beta+d)}.$$

Although only the case $\alpha = 1$ is considered in Samworth (2012), the convergence rate in Corollary 1 coincides with the rate for LP-classifier, given in Proposition 3.

Whereas theories can be constructed without solving the equations, solving the equations to determine the optimal real-valued weights explicitly is rather burdensome, where the explicit solution is shown only for $u = 1, 2$ (namely, $\beta = 2, 4$) in Samworth (2012); the solution for $u = 2$ is $a_1 := \frac{1}{(k_*)^{2/d}} \left\{ \frac{(d+4)^2}{4} - \frac{2(d+4)}{d+2} a_0 \right\}$, $a_2 = \frac{1 - a_0 - (k_*)^{2/d} a_1}{(k_*)^{4/d}}$ (see, Supp. D for more details, and also see Figure 2(a) in Supp. E for the optimal weights computed in an experiment ($u = 2$)).

We also note that, conducting cross-validation to choose $(w_1, w_2, \ldots, w_k)$ directly from $\mathcal{W}^k$ (for some sets $\mathcal{W} \subset \mathbb{R}$) is impractical, as it requires large computational cost $O(|\mathcal{W}|^k)$. Therefore, other simpler approaches to determine optimal real-valued weights are appreciated in practice.

# 4 Proposed multiscale $k$-NN

In this section, we propose multiscale $k$-NN (MS-$k$-NN), that implicitly finds favorable real-valued weights for weighted $k$-NN. Note that the obtained weights are different from Samworth (2012), as illustrated in Figure 2 in Supplement E. In what follows, we first formally define MS-$k$-NN in Section 4.1. Subsequently, the weights obtained via MS-$k$-NN are shown in Section 4.2, the convergence rate is discussed in Section 4.3.

## 4.1 Multiscale $k$-NN

**Underlying idea:** Since $\eta^{(\infty)}(B(X_*; r))$ asymptotically approximates the $k$-NN estimator $\hat{\eta}_{n,k}^{(k\mathrm{NN})}(X_*)$ for roughly $r = r(k) := \|X_{(k)} - X_*\|$ (see, e.g., Chaudhuri & Dasgupta (2014) Lemma 9), asymptotic expansion in Theorem 1 indicates that $\hat{\eta}_{n,k}^{(k\mathrm{NN})}(X_*) \approx \eta(X_*) + \sum_{c=1}^{\lfloor \beta/2 \rfloor} b_c^* r^{2c}$, for some $\{b_c^*\} \subset \mathbb{R}$. Estimating a function $f_{X_*}(r) := b_0 + \sum_{c=1}^{\lfloor \beta/2 \rfloor} b_c r^{2c}$ to predict the $k$-NN estimators for $k_1, k_2, \ldots, k_V$ and extrapolating to $k = 0$ via $r = r(k)$ with $r(0) := 0$ yields $\hat{f}_{X_*}(0) = \hat{b}_0 \approx \eta(X_*)$; the asymptotic bias $\sum b_c^* r^{2c}$ is then eradicated.

**Definition of MS-$k$-NN:** Let $V, C \in \mathbb{N}$ and fix any query $X_* \in \mathcal{X}$. We first compute unweighted $k$-NN estimators for $1 \leq k_1 < k_2 < \cdots < k_V \leq n$, i.e., $\hat{\eta}_{n,k_1}^{(k\mathrm{NN})}(X_*), \hat{\eta}_{n,k_2}^{(k\mathrm{NN})}(X_*), \ldots, \hat{\eta}_{n,k_V}^{(k\mathrm{NN})}(X_*)$. Then, we compute $r_v := \|X_{(k_v)} - X_*\|_2$, and consider a simple regression such that

$$\hat{\eta}_{n,k_v}^{(k\mathrm{NN})}(X_*) \approx b_0 + b_1 r_v^2 + b_2 r_v^4 + \cdots + b_C r_v^{2C} \tag{11}$$

for all $v \in [V]$, where $\boldsymbol{b} = (b_0, b_1, \ldots, b_C)$ is a regression coefficient vector to be estimated. Note that the regression function is a polynomial of $r_v^2$ which contains only terms of even degrees $r_v^{2c}$, since all the bias terms are of even degrees as shown in Theorem 1. However, it is certainly possible that we employ a polynomial with terms of odd degrees in practical cases.

More formally, we consider a minimization problem

$$\hat{\boldsymbol{b}} := \underset{\boldsymbol{b} \in \mathbb{R}^{C+1}}{\arg\min} \sum_{v=1}^{V} \left( \hat{\eta}_{n,k_v}^{(k\mathrm{NN})}(X_*) - b_0 - \sum_{c=1}^{C} b_c r_v^{2c} \right)^2. \tag{12}$$

Then, we propose a *multiscale k-NN (MS-k-NN)* estimator

$$\hat{\eta}_{n,\boldsymbol{k}}^{(\text{MS-}k\text{NN})}(X_*) := \hat{b}_0 \quad \left(= \boldsymbol{z}(X_*)^\top \hat{\boldsymbol{\eta}}_{n,\boldsymbol{k}}^{(k\text{NN})}(X_*)\right), \tag{13}$$

where $\boldsymbol{k} = (k_1, k_2, \ldots, k_V) \in \mathbb{N}^V$, $\hat{\boldsymbol{\eta}}_{n,\boldsymbol{k}}^{(k\text{NN})}(\boldsymbol{X}_*) := (\hat{\eta}_{n,k_1}^{(k\text{NN})}(X_*), \ldots, \hat{\eta}_{n,k_V}^{(k\text{NN})}(X_*))^\top \in \mathbb{R}^V$ and $\boldsymbol{z}(X_*) \in \mathbb{R}^V$ will be defined in (15). Since (11) extrapolates $k$-NN estimators to $r = 0$, we also call the situation by "extrapolating to $k = 0$" analogously. The corresponding *MS-k-NN classifier* is defined as

$$\hat{g}_{n,\boldsymbol{k}}^{(\text{MS-}k\text{NN})}(X) := g^{(\text{plug-in})}(X; \hat{\eta}_{n,\boldsymbol{k}}^{(\text{MS-}k\text{NN})}). \tag{14}$$

Note that the number of terms in the regression function (11) is $1 + C$, and $C$ will be specified as $C = \lfloor \beta/2 \rfloor$ under the $\beta$-Hölder condition in Theorem 2. Although the parameter $\beta$ cannot be observed in practice, we may employ large $C$ so that $C \geq \lfloor \beta/2 \rfloor$ is expected (e.g., $C = 10$). Even in this case, overall number of terms in the MS-$k$-NN is $1 + C$, which is much less than the number of coefficients used in the LP classifier $(= 1 + d + d^2 + \cdots + d^C)$.

## 4.2 Corresponding real-valued weights

In this section, real-valued weights implicitly obtained via MS-$k$-NN are considered. The vector $\boldsymbol{z}(X_*) = (z_1(X_*), z_2(X_*), \ldots, z_V(X_*))^\top \in \mathbb{R}^V$ in the definition of MS-$k$-NN (13) is obtained by simply solving the minimization problem (12), as

$$\boldsymbol{z}(X_*) := \frac{(\boldsymbol{I} - \mathcal{P}_{\boldsymbol{R}(X_*)})\mathbf{1}}{V - \mathbf{1}^\top \mathcal{P}_{\boldsymbol{R}(X_*)}\mathbf{1}}, \tag{15}$$

where $\mathbf{1} = (1, 1, \ldots, 1)^\top \in \mathbb{R}^V$, $\mathcal{P}_{\boldsymbol{R}} = \boldsymbol{R}(\boldsymbol{R}^\top \boldsymbol{R})^{-1}\boldsymbol{R}$ and $(i, j)$-th entry of $\boldsymbol{R} = \boldsymbol{R}(X_*)$ is $r_i^{2j}$ for $(i, j) \in [V] \times [C]$; note that the radius $r_i$ depends on the query $X_*$. Therefore, the corresponding optimal real-valued weight $\boldsymbol{w}_*(X_*) = (w_1^*(X_*), w_2^*(X_*), \ldots, w_{k_V}^*(X_*))$ is obtained as

$$w_i^*(X_*) := \sum_{v:i \leq k_v} \frac{z_v(X_*)}{k_v} \in \mathbb{R}, \quad (\forall i \in [k_V]), \tag{16}$$

then $\hat{\eta}_{n,k_V,\boldsymbol{w}_*}^{(k\text{NN})}(X_*) = \hat{\eta}_{n,\boldsymbol{k}}^{(\text{MS-}k\text{NN})}(X_*)$. Here, we note that the weight (16) is adaptive to the query $X_*$, as each entry of the matrix $\boldsymbol{R}$ used in the definition of $\boldsymbol{z}$ (15) depends on both sample $\mathcal{D}_n$ and query $X_*$. See Supplement E for the skipped derivation of the above (15) and (16). Total sum of the weights (16) is then easily proved as $\sum_{i=1}^{k_V} w_i^*(X_*) = \sum_{v=1}^V z_v(X_*) = \mathbf{1}^\top \boldsymbol{z}(X_*) \stackrel{(15)}{=} 1$.

To give an example, we plotted the weights in the following Figure 2. The real-valued weights implicitly computed in MS-$k$-NN are plotted to compare with the optimal real-valued weights proposed by Samworth (2012).

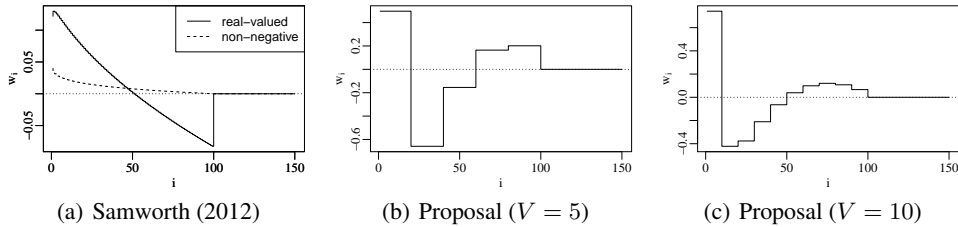

|        (a) Samworth (2012)        |        (b) Proposal ($V = 5$)        |        (c) Proposal ($V = 10$)        |

Figure 2: Amongst all the experiments, $n = 1000, d = 10, u = C = 2, k_* = 100$. In (a), optimal non-negative (8) and real-valued (10) weights for weighted $k$-NN (7) in Samworth (2012) are plotted. In (b) and (c), real-valued weights (16) implicitly computed in the **proposed MS-$k$-NN**, are plotted for $k_v = k_* v/V, r_v := (k_v/n)^{1/d}$ $(v \in [V])$. $V$ is the number of $k$ used for regression.

Figure 2(b) and 2(c) illustrate the optimal weights (16) for $V = 5, 10$. The weights are not monotonically decreasing for $i \leq k_V (= 100)$, and the weights are smoothly connected to $w_{k_V+1}^* = 0$ at $i \approx k_V$, unlike Samworth (2012) shown in Figure 2(a).

Although the weights (16) can be easily computed, they are not computed in practice. Only a procedure needed for MS-$k$-NN is to conduct the regression (12) and specify $\hat{\eta}_{n,k}^{(\text{MS-}k\text{NN})}$ by the intercept $\hat{b}_0$ stored in the regression coefficient $\hat{\boldsymbol{b}}$. Then, MS-$k$-NN automatically coincides with the weighted $k$-NN using the above optimal weight (16).

### 4.3 Convergence rate for MS-$k$-NN classifier

Here, we consider the convergence rate for MS-$k$-NN classifier. Firstly, we specify a vector $\boldsymbol{\ell} = (\ell_1, \ell_2, \ldots, \ell_V) \in \mathbb{R}^V$ so that $\ell_1 = 1 < \ell_2 < \cdots < \ell_V < \infty$. We assume that

(C-1) $k_{1,n} \asymp n^{2\beta/(2\beta+d)}$,

(C-2) $k_{v,n} := \min\{k \in [n] \mid \|X_{(k)} - X_*\|_2 \geq \ell_v r_{1,n}\}$ for $v = 2, 3, \ldots, V$, where $r_{1,n} := \|X_{(k_{1,n})} - X_*\|_2$,

(C-3) $\exists L_{\boldsymbol{z}} > 0$ such that $\|\boldsymbol{z}_{\boldsymbol{\ell}}\|_\infty < L_{\boldsymbol{z}}$, where $\boldsymbol{z}_{\boldsymbol{\ell}} = \frac{(\boldsymbol{I} - \mathcal{P}_{\boldsymbol{R}})\boldsymbol{1}}{\boldsymbol{1}^\top(\boldsymbol{I} - \mathcal{P}_{\boldsymbol{R}})\boldsymbol{1}}$ and $\boldsymbol{R} = (\ell_i^{2j})_{ij} \in \mathbb{R}^{[V] \times [C]}$, for all $X_* \in \mathcal{S}(\mu)$.

Intuition behind the above conditions (C-1)–(C-3) is as follows. (C-1): local polynomial classifier with bandwidth $h = h_n \asymp n^{-1/(2\beta+d)}$ is known to attain the optimal rate (Audibert & Tsybakov, 2007, Theorem 3.3). Therefore, the information in the ball $B(X; h)$ with radius $h > 0$ is roughly required for the optimal rate. When assuming that the feature vectors $X_1, X_2, \ldots, X_n$ distribute uniformly, $k$ and the bandwidth $h$ have the relation $k \asymp nh^d \asymp n^{2\beta/(2\beta+d)}$ as the volume of $B(X; h)$ is of order $h^d$. (C-2): $k_{1,n}, k_{2,n}, \ldots, k_{v,n}$ are selected so that $r_{2,n}, \ldots, r_{v,n}$ are in the same order as $r := r_{1,n}$ (and $r_{v,n}/r_{1,n} \to \ell_v$). (C-3): the weights $w_1, w_2, \ldots, w_k$ estimated via regression take finite values asymptotically.

Then, regarding the MS-$k$-NN estimator (13) and its corresponding MS-$k$-NN classifier (14), the following Theorem 2 holds.

**Theorem 2** (Convergence rate for MS-$k$-NN). Assuming that (i) $\mu$ and $\eta\mu$ are $\beta$-Hölder, (ii) $\mu$ satisfies the strong density asumption, (iii) $C := \lfloor \beta/2 \rfloor \leq V - 1$, and (iv) the conditions (C-1)–(C-3) are satisfied. Then,

$$\mathcal{E}(\hat{g}_{n,\boldsymbol{k}}^{(\text{MS-}k\text{NN})}) = O(n^{-(1+\alpha)\beta/(2\beta+d)}).$$

*Sketch of Proof:* By following the underlying idea explained at first in Section 4.1, the bias $O(r^{\min\{2,\beta\}})$ of the conventional $k$-NN is reduced to $O(r^\beta)$. Therefore, intuitively speaking, replacing the bias term in the proof of Chaudhuri & Dasgupta (2014) Theorem 4(b) leads to the proof of Theorem 2. Although this proof stands on the simple underlying idea, technically speaking, some additional considerations are needed; see Supplement G for detailed proof. □

The rate obtained in Theorem 2 coincides with the optimal rate provided in Corollary 1; MS-$k$-NN is an optimal classifier, at least for the case $\alpha = 1, \beta = 2u$ ($u \in \mathbb{N}$).

### 4.4 Using $\log k$ as the predictor

The standard MS-$k$-NN predicts unweighted $k$-NN estimators through the radius $r = r(k)$, that is computed via sample $\mathcal{D}_n$. As an alternative approach, we instead consider predicting the estimators directly from $k$.

For clarifying the relation between the radius $r = r(k)$ and $k$, we here consider the simplest setting that the feature vector $X$ distributes uniformly. Then, $r_v := \|X_{(k_v)} - X_*\|_2$ used in (11) is roughly proportional to $k_v^{1/d}$ since the volume of the ball of radius $r_v$ is proportional to $r_v^d$.

Then, for sufficiently large $d$,

$$r_v^2 \propto k_v^{2/d} = \exp\left(\frac{2}{d} \log k_v\right) = 1 + \frac{2}{d} \log k_v + O(d^{-2}). \tag{17}$$

Thus, (11) can be expressed as a polynomial with respect to $\log k_v$ instead of $r_v^2$. In numerical experiments, we then extrapolate unweighted $k$-NN to $k = 1$.

# 5 Numerical experiments

**Datasets:** We employ datasets from UCI Machine Learning Repository (Dua & Graff, 2017). Each of datasets consists of $d$-dimensional $n$ feature vectors $X_i \in \mathcal{X}$, and their labels $Y_i \in \{1, 2, \ldots, m\}$ representing 1 of $m$ categories.

**Preprocessing:** Feature vectors are first normalized, and then randomly divided into 70% for prediction ($n_{\text{pred}} = \lfloor 0.7n \rfloor$) and the remaining for test query.

**Evaluation metric:** Category of the query is predicted so that the corresponding estimator attains the maximum value. The classification accuracy is evaluated via 10 times experiments. The MS-$k$-NN estimated via radius $r(k)$ and that via $\log k$ described in Section 4.4 are performed with $C = 1$; they are compared with unweighted $k$-NN and weighted $k$-NN with the optimal non-negative and real-valued weights (Samworth, 2012). Regression in MS-$k$-NN is ridge regularized with the coefficient $\lambda = 10^{-4}$.

**Parameter tuning:** For unweighted and weighted $k$-NN, we first fix $k := V \cdot \lfloor n_{\text{pred}}^{4/(4+d)} \rfloor \asymp n_{\text{pred}}^{4/(4+d)}$. Using the same $k$, we simply choose $k_1 := k/V, k_2 = 2k/V, \ldots, k_V = k$ with $V = 5$ for MS-$k$-NN.

**Results:** Sample mean and the sample standard deviation on 10 experiments are shown in Table 2. Overall, weighted $k$-NN and MS-$k$-NN show better score than unweighted $k$-NN ($w_i = 1/k$). MS-$k$-NN via radius $r(k)$ shows the best or second best score for (all of) 13 datasets; this number is maximum among all the methods considered in these experiments. As well as the MS-$k$-NN using $r(k)$, MS-$k$-NN using the alternative predictor $\log k$ also shows promising scores for some datasets. Regarding larger datasets such as MAGIC and Avila, weighted $k$-NN equipped with real-valued weights, which are computed by both ways of Samworth (2012) and MS-$k$-NN, demonstrate slightly better performance than the weighted $k$-NN with non-negative weights; this observation coincides with the theoretical optimality.

Table 2: Each dataset consists of $n$ feature vectors whose dimension is $d$; each object is labeled by 1 of $m$ categories. Sample average and the standard deviation for the prediction accuracy are computed on 10 times experiments. Best scores are **bolded**, and second best scores are underlined.

| Dataset | $n$ | $d$ | $m$ | $k$-NN | | | MS-$k$-NN | |
|---|---|---|---|---|---|---|---|---|
| | | | | $w_i = 1/k$ | $w_i \geq 0$ (8) | $w_i \in \mathbb{R}$ (10) | via $r(k)$ (13) | via $\log k$ (17) |
| Iris | 150 | 4 | 3 | $0.83 \pm 0.04$ | $0.92 \pm 0.05$ | $0.92 \pm 0.04$ | $0.93 \pm 0.04$ | **$0.96 \pm 0.04$** |
| Glass identification | 213 | 9 | 6 | $0.58 \pm 0.06$ | $0.64 \pm 0.06$ | **$0.67 \pm 0.05$** | $0.64 \pm 0.05$ | $0.64 \pm 0.05$ |
| Ecoli | 335 | 7 | 8 | $0.80 \pm 0.03$ | **$0.85 \pm 0.03$** | $0.84 \pm 0.02$ | **$0.85 \pm 0.02$** | $0.84 \pm 0.02$ |
| Diabetes | 768 | 8 | 2 | **$0.75 \pm 0.03$** | $0.74 \pm 0.03$ | $0.70 \pm 0.04$ | **$0.75 \pm 0.03$** | $0.71 \pm 0.03$ |
| Biodegradation | 1054 | 41 | 2 | $0.84 \pm 0.02$ | **$0.86 \pm 0.03$** | $0.79 \pm 0.02$ | **$0.86 \pm 0.02$** | $0.80 \pm 0.02$ |
| Banknote | 1371 | 4 | 2 | $0.95 \pm 0.01$ | $0.98 \pm 0.01$ | $0.97 \pm 0.01$ | $0.98 \pm 0.01$ | **$0.99 \pm 0.00$** |
| Yeast | 1484 | 8 | 10 | $0.57 \pm 0.02$ | **$0.58 \pm 0.02$** | $0.54 \pm 0.03$ | **$0.58 \pm 0.02$** | $0.54 \pm 0.02$ |
| Wireless localization | 2000 | 7 | 4 | $0.97 \pm 0.00$ | **$0.98 \pm 0.00$** | **$0.98 \pm 0.01$** | **$0.98 \pm 0.00$** | **$0.98 \pm 0.01$** |
| Spambase | 4600 | 57 | 2 | $0.90 \pm 0.01$ | **$0.91 \pm 0.00$** | $0.86 \pm 0.01$ | **$0.91 \pm 0.00$** | $0.87 \pm 0.01$ |
| Robot navigation | 5455 | 24 | 4 | $0.81 \pm 0.01$ | **$0.86 \pm 0.01$** | $0.81 \pm 0.01$ | $0.84 \pm 0.01$ | $0.84 \pm 0.01$ |
| Page blocks | 5473 | 10 | 5 | $0.95 \pm 0.01$ | $0.95 \pm 0.01$ | **$0.96 \pm 0.01$** | **$0.96 \pm 0.01$** | **$0.96 \pm 0.01$** |
| MAGIC | 19020 | 10 | 2 | $0.82 \pm 0.00$ | $0.82 \pm 0.00$ | **$0.84 \pm 0.01$** | $0.83 \pm 0.00$ | $0.83 \pm 0.00$ |
| Avila | 20867 | 10 | 12 | $0.63 \pm 0.01$ | $0.68 \pm 0.01$ | **$0.70 \pm 0.01$** | $0.69 \pm 0.00$ | **$0.70 \pm 0.01$** |

# 6 Conclusion and future works

In this paper, we proposed multiscale $k$-NN (13), that extrapolates $k$-NN estimators from $k \geq 1$ to $k = 0$ via regression. MS-$k$-NN corresponds to finding favorable real-valued weights (16) for weighted $k$-NN, and it attains the convergence rate $O(n^{-(1+\alpha)\beta/(2\beta+d)})$ shown in Theorem 2. It coincides with the optimal rate shown in Samworth (2012) in the case $\alpha = 1, \beta = 2u$ ($u \in \mathbb{N}$). For future work, it would be worthwhile to relax assumptions in theorems, especially the $\beta$-Hölder condition on $\mu$ and the limitation on the distance to Euclidean. As also noted at last in Samworth (2012) Section 4, rather larger sample sizes would be needed for receiving benefits from asymptotic theories; adaptation to small samples and high-dimensional settings are also appreciated.

## Broader Impact

For improving the convergence rate of the conventional $k$-NN, we propose to consider a simple extrapolation idea; it provides an intuitive understanding of not only the optimal $k$-NN but also more general nonparametric statistics. By virtue of the simplicity, MS-$k$-NN is also easy to implement; the similar idea may be applied to some other statistical and machine learning methods.

## Acknowledgement

We would like to thank Ruixing Cao and Takuma Tanaka for helpful discussions. This work was partially supported by JSPS KAKENHI grant 16H02789, 20H04148 to HS.

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
