[Supplementary Material]

# Supplementary Material:

## Extrapolation Towards Imaginary 0-Nearest Neighbour and Its Improved Convergence Rate

## A Related works

Györfi (1981) is the first work that proves the convergence rate $O(n^{-\gamma/(2\gamma+d)})$ for unweighted $k$-NN classifier by assuming the $\gamma$-neighbour average smoothness, and the rate is improved by Chaudhuri & Dasgupta (2014), by additionally imposing the $\alpha$-margin condition.

For choosing adaptive $k = k(X_*)$ with non-negative weights $w_i = 1/k$, i.e., $k$ depending on the query $X_*$, Balsubramani et al. (2019) considers the confidence interval of the $k$-NN estimator from the decision boundary, and Cannings et al. (2017) considers the asymptotic expansion used in Samworth (2012) and obtains the rate of $O(n^{-4/(4+d)})$, same rate as unweighted $k$-NN up to constant factor. Anava & Levy (2016) considers adaptive non-negative weights and $k = k(X_*)$ but the approach is rather heuristic.

## B Other classifiers and their convergence rates

In this section, we describe Nadaraya-Watson (NW) classifier, Local Polynomial (LP) classifier and their convergence rates (Audibert & Tsybakov, 2007). In what follows, $K : \mathcal{X} \to \mathbb{R}$ represents a kernel function, e.g., Gaussian kernel $K(X) := \exp(-\|X\|_2^2)$, and $h > 0$ represents a bandwidth.

**Definition 6.** *Nadaraya-Watson (NW) estimator* is defined as

$$\hat{\eta}_{n,h}^{(\text{NW})}(X_*) := \frac{\sum_{i=1}^{n} Y_i \, K\left(\frac{X_i - X_*}{h}\right)}{\sum_{i=1}^{n} K\left(\frac{X_i - X_*}{h}\right)}$$

if the denominator is nonzero, and it is zero otherwise. $\hat{g}_{n,h}^{(\text{NW})}(X) := g^{(\text{plug-in})}(X; \hat{\eta}_{n,h}^{(\text{NW})})$ is called *NW-classifier*.

Here, we define a loss function

$$\mathcal{L}_{n,h}(f, X_*) := \sum_{i=1}^{n} \{Y_i - f(X_i - X_*)\}^2 \, K\left(\frac{X_i - X_*}{h}\right) \tag{18}$$

for $f : \mathcal{X} \to \mathbb{R}$; Using a constant function $f(x) \equiv \theta$, NW estimator can be regarded as a minimizer $\theta \in \mathbb{R}$ of $(\mathcal{L}_{n,h}(f, X_*))$. NW estimator is then generalized to the local polynomial (LP) estimator when $Y_i$ is predicted by a polynomial function.

**Definition 7.** Let $\mathcal{F}_q$ denotes the set of polynomial functions $f : \mathcal{X} \to \mathbb{R}$ of degree $q \in \mathbb{N}_0$. Considering the function

$$\hat{f}_{n,h,q}^{X_*} := \underset{f \in \mathcal{F}_q}{\arg\min} \, \mathcal{L}_{n,h}(f, X_*), \tag{19}$$

*local polynomial (LP) estimator* of degree $q$ is defined as $\hat{\eta}_{n,h,q}^{(\text{LP})}(X_*) := \hat{f}_{n,h,q}^{X_*}(\mathbf{0})$ if $\hat{f}_{n,h,q}^{X_*}$ is the unique minimizer of $\mathcal{L}_{n,h}(f, X_*)$ and it is zero otherwise. The corresponding $\hat{g}_{n,h,q}^{(\text{LP})}(X) := g^{(\text{plug-in})}(X; \hat{\eta}_{n,h,q}^{(\text{LP})})$ is called *LP classifier*.

Note that LP classifier is computed via polynomial function of degree $q$; they contain $1 + d + d^2 + \cdots + d^q$ terms therein, and it results in high computational cost if $d, q$ are large.

**Proposition 3** (Audibert & Tsybakov (2007) Th. 3.3). Let $\mathcal{X}$ be a compact set, and assuming that (i) $\eta$ satisfies $\alpha$-margin condition and is $\beta$-Hölder, and (ii) $\mu$ satisfies strong densitiy assumption. Then, the convergence rate of the LP classifier with the bandwidth $h_* = h_n \asymp n^{-1/(2\beta+d)}$ is

$$\mathcal{E}(\hat{g}_{n,h_*,\lfloor\beta\rfloor}^{(\text{LP})}) = O(n^{-(1+\alpha)\beta/(2\beta+d)}).$$

The above Proposition 3 indicates that, the convergence rate for the LP classifier is faster than $O(n^{-1/2})$ for $\alpha\beta > d/2$, and the rate is even faster than $O(n^{-1})$ for $(\alpha - 1)\beta > d$, though such inequalities are rarely satisfied since the dimension $d$ is large in many practical situations.

Rigorously speaking, Audibert & Tsybakov (2007) considers the uniform bound of the excess risk over all the possible $(\eta, \mu)$, and Audibert & Tsybakov (2007) Theorem 3.5 proves the optimality of the rate, i.e., $\sup_{(\eta,\mu)} \mathcal{E}(g) \geq \exists C \cdot n^{-(1+\alpha)\beta/(2\beta+d)}$

for any classifier $g$ when $\alpha\beta < d$. LP classifier is thus proved to be an optimal classifier in this sense. However, the optimality is for uniform evaluation $\sup_{(\eta,\mu)} \mathcal{E}(\cdot)$, but not the non-uniform evaluation $\mathcal{E}(\cdot)$, that is considered in this paper; it remains unclear whether the (non-uniform) evaluation is still lower-bounded by $n^{-(1+\alpha)\beta/(2\beta+d)}$ if sup is removed. In particular, the uniform bound of NW classifier (i.e., LP classifier with $\lfloor\beta\rfloor = 0$) is $O(n^{-2/(2+d)})$ for $\alpha = \beta = 1$, but it is slower than the convergence rate $O(n^{-4/(4+d)})$ of NW classifier.

We last note that the LP classifier leverages the polynomial of degree $q$, that is defined in Definition 7; it contains $1 + d + d^2 + \cdots + d^q$ terms, resulting in high computational cost as the dimension $d$ of feature vectors is usually not that small.

## C  A Note on Proposition 1

Regarding the symbols, $(\alpha, \beta)$ in Chaudhuri & Dasgupta (2014) correspond to $(\tilde{\gamma}, \alpha)$ in this paper, where $\tilde{\gamma} := \gamma/d$ is formally defined in the following. Chaudhuri & Dasgupta (2014) in fact employs "$(\alpha, L)$-smooth" condition

$$|\eta(X_*) - \eta^{(\infty)}(B(X_*; r))| \le L \left( \int_{B(X_*;r)} \mu(X)\mathrm{d}X \right)^{\tilde{\gamma}}, \tag{20}$$

which is different from our definition of the $\gamma$-neighbour average smoothness, i.e.,

$$|\eta(X_*) - \eta^{(\infty)}(B(X_*; r))| \le L_\gamma r^\gamma. \tag{21}$$

However, their definition (20) can be obtained from our definition (21), by imposing an additional assumption $\mu(X) \ge \mu_{\min}$ for all $X \in \mathcal{X}$. The proof is straightforward: the integrant in (20) is lower-bounded by

$$\int_{B(X_*;r)} \mu(X)\mathrm{d}X \ge \mu_{\min} \frac{\pi^{d/2}}{\Gamma(1+d/2)} r^d =: Dr^d,$$

then

$$|\eta(X_*) - \eta^{(\infty)}(B(X_*; r))| \overset{(21)}{\le} L_\gamma r^\gamma \le L_\gamma \left( \frac{1}{D} \int_{B(X_*;r)} \mu(X)\mathrm{d}X \right)^{\gamma/d} = L \left( \int_{B(X_*;r)} \mu(X)\mathrm{d}X \right)^{\tilde{\gamma}}$$

by specifying $L := L_\gamma/D^{\gamma/d}, \tilde{\gamma} = \gamma/d$. Therefore, Chaudhuri & Dasgupta (2014) Theorem 4(b) proves Proposition 1, by considering the above correspondence of the symbols and the assumption.

## D  Samworth (2012) Theorem 6

For each $s \in (0, 1/2)$, $\mathcal{W}_{n,s}$ denotes the set of all sequences of real-valued weight vectors $\boldsymbol{w}_n := (w_1, w_2, \ldots, w_n) \in \mathbb{R}^n$ satisfying

$$\sum_{i=1}^n w_i = 1, \quad \frac{n^{2u/d} \sum_{i=1}^n \delta_i^{(\ell)} w_i}{n^{2\ell/d} \sum_{i=1}^n \delta_i^{(u)} w_i} \le \frac{1}{\log n} \quad (\forall \ell \in [u-1]),$$

$$\sum_{i=1}^n w_i^2 \le n^{-s},$$

$$n^{-4u/d} (\sum_{i=1}^n \delta_i^{(u)} w_i)^2 \le n^{-s},$$

$$\exists k_2 \le \lfloor n^{1-s} \rfloor \text{ s.t. } \frac{n^{2u/d} \sum_{i=k_2+1}^n |w_i|}{\sum_{i=1}^n \delta_i^{(u)} w_i} \le \frac{1}{\log n} \text{ and } \sum_{i=1}^{k_2} \delta_i^{(u)} w_i \ge \beta k_2^{2u/d},$$

$$\frac{\sum_{i=k_2+1}^n w_i^2}{\sum_{i=1}^n w_i^2} \le \frac{1}{\log n},$$

$$\frac{\sum_{i=1}^n |w_i|^3}{(\sum_{i=1}^n w_i^2)^{3/2}} \le \frac{1}{\log n},$$

where $\delta_i^{(\ell)} := i^{1+2\ell/d} - (i-1)^{1+2\ell/d}$ for all $\ell \in [u-1]$.

For the rigorous proof, Samworth (2012) considers the following assumptions.

(i) $\mathcal{X} \subset \mathbb{R}^d$ is a compact $d$-dimensional manifold with boundary $\partial \mathcal{X}$,

(ii) $\mathcal{S} := \{x \in \mathcal{X} \mid \eta(x) = 1/2\}$ is nonempty. There exists an open subset $U_0 \subset \mathbb{R}^d$ that contains $\mathcal{S}$ and such that the following properties hold: (1) $\eta$ is continuous on $U \setminus U_0$, where $U$ is an open set containing $\mathcal{X}$, (2) restrictions of $P_0(X) := \mathbb{P}(X \mid Y = 0), P_1(X) := \mathbb{P}(X \mid Y = 1)$ to $U_0$ are absolutely continuous w.r.t. Lebesgue measure, with $2u$-times continuously differentiable ($C^{2u}$) Radon-Nikodym derivatives $f_0, f_1$, respectively. Since $f_0, f_1 \in C^{2u}$, we also have $\eta(x) = \mathbb{P}(Y = 1)f_1(x)/(\mathbb{P}(Y = 0)f_0(x) + \mathbb{P}(Y = 1)f_1(x))$ is $C^{2u}$.

(iii) There exists $\rho > 0$ such that $\int_{\mathbb{R}^d} \|x\|^\rho \mathrm{d}\mathbb{P}(x) < \infty$. Moreover, for sufficiently small $r > 0$, the ratio $\mathbb{P}(B(x; r))/(a_d r^d)$ is bounded away from zero, uniformly for $x \in \mathcal{X}$.

(iv) $\partial \eta(x)/\partial x \neq 0$ for all $x \in \mathcal{X}$ and its restriction to $\mathcal{S}$ is also nonzero for all $x \in \mathcal{S} \cap \partial \mathcal{X}$.

**Proposition 4** (Samworth (2012) Theorem 6). Assuming that (i)–(iv), it holds for each $s \in (0, 1/2)$ that

$$\mathcal{E}(\hat{g}_{n,k,\boldsymbol{w}}^{(k\mathrm{NN})}) = \underbrace{\left( B_1 \sum_{i=1}^n w_i^2 + B_2 \left( \sum_{i=1}^n \frac{\delta_i^{(u)} w_i}{n^{2u/d}} \right)^2 \right)}_{=:\gamma_n(\boldsymbol{w}_n)} \{1 + o(1)\}$$

for some constants $B_1, B_2 > 0$, as $n \to \infty$, uniformly for $\boldsymbol{w} \in \mathcal{W}_{n,s}$, and $\delta_i^{(\ell)} := i^{1+2\ell/d} - (i-1)^{1+2\ell/d}, \ell \in [u-1]$.

Whereas the weights are constrained as

$$\sum_{i=1}^n w_i = 1, \quad \sum_{i=1}^n \delta_i^{(\ell)} w_i = 0 \quad (\forall \ell \in [u-1]), \tag{22}$$

and $w_i = 0$ for $i = k^* + 1, \ldots, n$ with $k^* \asymp n^{2\beta/(2\beta+d)}$. Samworth (2012) eq. (4.3) shows that the optimal weight should be in the form

$$w_i^* := \begin{cases} (a_0 + a_1 \delta_1^{(i)} + \cdots + a_u \delta_i^{(u)})/k^* & (i \in [k^*]) \\ 0 & (\text{otherwise.}) \end{cases} \tag{23}$$

Coefficients $\boldsymbol{a} = (a_0, a_1, \ldots, a_u)$ are determined by solving the equations (22) and (23) simultaneously; then the optimal weights are obtained by substituting it to (23).

They also show the asymptotic solution of the above equations, in the case of $u = 2$; the solution is

$$a_1 = \frac{1}{(k^*)^{2/d}} \left\{ \frac{(d+4)^2}{4} - \frac{2(d+4)}{d+2} a_0 \right\}, \quad a_2 = \frac{1 - a_0 - (k^*)^{2/d} a_1}{(k^*)^{4/d}}.$$

# E   Real-valued Weights Obtained via MS-$k$-NN

Let $X_* \in \mathcal{X}$ any given query, and let denote $k$-NN estimator by $\varphi_{n,k} := \hat{\eta}_{n,k}^{(k\mathrm{NN})}(X_*)$. Considering

$$\boldsymbol{\varphi}_{n,\boldsymbol{k}} = \boldsymbol{\varphi}_{n,\boldsymbol{k}}(X_*) := (\varphi_{n,k_1}, \varphi_{n,k_2}, \ldots, \varphi_{n,k_V}) \in \mathbb{R}^V, \tag{24}$$

$$\boldsymbol{R} = \boldsymbol{R}(X_*) := \begin{pmatrix} r_1^2 & r_1^4 & \cdots & r_1^{2C} \\ r_2^2 & r_2^4 & \cdots & r_2^{2C} \\ \vdots & \vdots & \ddots & \vdots \\ r_V^2 & ,r_V^4 & \cdots & r_V^{2C} \end{pmatrix} \in \mathbb{R}^{V \times C}, \tag{25}$$

$$\boldsymbol{A} = \boldsymbol{A}(X_*) := (\boldsymbol{1}\ \boldsymbol{R}) \in \mathbb{R}^{V \times (C+1)}, \tag{26}$$

$$\boldsymbol{b} = \boldsymbol{b}(X_*) := (b_0, b_1, b_2, \ldots, b_C) \in \mathbb{R}^{C+1}, \tag{27}$$

the minimization problem (12) becomes

$$\hat{\boldsymbol{b}} = \underset{\boldsymbol{b} \in \mathbb{R}^{C+1}}{\arg\min} \sum_{v=1}^V \left( \hat{\eta}_{n,k}^{(k\mathrm{NN})}(X_*) - \sum_{c=0}^C b_c r_v^{2c} \right) = \underset{\boldsymbol{b} \in \mathbb{R}^{C+1}}{\arg\min} \|\boldsymbol{\varphi}_{\boldsymbol{k},n} - \boldsymbol{A}\boldsymbol{b}\|_2^2 = \underbrace{(\boldsymbol{A}^\top \boldsymbol{A})^{-1} \boldsymbol{A}^\top}_{(\star)} \boldsymbol{\varphi}_{\boldsymbol{k},n}.$$

Therefore, denoting the first row of the matrix $(\star)$ by the vector $\boldsymbol{z}^\top = (z_1, z_2, \ldots, z_V)^\top \in \mathbb{R}^V$, MS-$k$-NN estimator is $\hat{\eta}_{\boldsymbol{k},n}^{\mathrm{MS}\text{-}k\mathrm{NN}}(X_*) = \hat{b}_0 = \boldsymbol{z}^\top \boldsymbol{\varphi}_{\boldsymbol{k},n}$. To obtain the explicit form of $\boldsymbol{z}$, we hereafter expand the matrix $(\star)$.

Considering the inverse of block matrix

$$
\begin{pmatrix} A & B \\ C & D \end{pmatrix}^{-1} = \begin{pmatrix} (A - BD^{-1}C)^{-1} & -(A - BD^{-1}C)^{-1}BD^{-1} \\ -D^{-1}C(A - BD^{-1}C)^{-1} & D^{-1} + D^{-1}C(A - BD^{-1}C)^{-1}BD^{-1} \end{pmatrix}
$$

(see, e.g., Petersen & Pedersen (2012) Section 9.1.3.), we have

$$
\begin{aligned}
(\star) &= \begin{pmatrix} V & \mathbf{1}^\top \boldsymbol{R} \\ \boldsymbol{R}^\top \mathbf{1} & \boldsymbol{R}^\top \boldsymbol{R} \end{pmatrix}^{-1} (\mathbf{1}\ \boldsymbol{R})^\top \\
&= \begin{pmatrix} \frac{1}{e} & -\frac{1}{e}\mathbf{1}^\top \boldsymbol{R}(\boldsymbol{R}^\top \boldsymbol{R})^{-1} \\ -\frac{1}{e}(\boldsymbol{R}^\top \boldsymbol{R}^{-1})^{-1}\boldsymbol{R}^\top \mathbf{1} & (\boldsymbol{R}^\top \boldsymbol{R})^{-1} + \frac{1}{e}(\boldsymbol{R}^\top \boldsymbol{R})^{-1}\boldsymbol{R}^\top \mathbf{1}\mathbf{1}^\top \boldsymbol{R}(\boldsymbol{R}^\top \boldsymbol{R})^{-1} \end{pmatrix} (\mathbf{1}\ \boldsymbol{R})^\top, \quad \text{where} \\
e &:= A - BD^{-1}C = V - \mathbf{1}^\top \boldsymbol{R}(\boldsymbol{R}^\top \boldsymbol{R})^{-1}\boldsymbol{R}^\top \mathbf{1} \in \mathbb{R}.
\end{aligned}
$$

Therefore, its first column is,

$$
\boldsymbol{z} = \frac{1}{e}\left\{ \boldsymbol{I} - \boldsymbol{R}(\boldsymbol{R}^\top \boldsymbol{R})^{-1}\boldsymbol{R}^\top \right\}\mathbf{1} = \frac{1}{V - \mathbf{1}^\top \boldsymbol{R}(\boldsymbol{R}^\top \boldsymbol{R})^{-1}\boldsymbol{R}^\top \mathbf{1}}\left\{ \boldsymbol{I} - \boldsymbol{R}(\boldsymbol{R}^\top \boldsymbol{R})^{-1}\boldsymbol{R}^\top \right\}\mathbf{1} = \frac{(\boldsymbol{I} - \mathcal{P}_{\boldsymbol{R}})\mathbf{1}}{V - \mathbf{1}^\top \mathcal{P}_{\boldsymbol{R}}\mathbf{1}},
$$

where $\mathcal{P}_{\boldsymbol{R}} := \boldsymbol{R}(\boldsymbol{R}^\top \boldsymbol{R})^{-1}\boldsymbol{R}^\top$ represents a projection matrix; the equation (15) is proved.

In addition, using the vector $\boldsymbol{z}$,

$$
\hat{\eta}_{\boldsymbol{k},n}^{\text{MS-}k\text{NN}}(X_*) = \boldsymbol{z}^\top \boldsymbol{\varphi}_{\boldsymbol{k},n} = \sum_{v=1}^{V} z_v \hat{\eta}_{k,n}^{(k\text{NN})} = \sum_{v=1}^{V} z_v \frac{1}{k_v} \sum_{i=1}^{k_v} Y_{(i)} = \sum_{i=1}^{k_V} w_i^* Y_{(i)} = \hat{\eta}_{n,k_V,\boldsymbol{w}_*}^{(k\text{NN})}(X_*),
$$

where

$$
w_i^* := \sum_{v:i \leq k_v} \frac{z_v}{k_v} \in \mathbb{R}, \quad (\forall i \in [k_V]),
$$

is the real-valued weight obtained via MS-$k$-NN. Thus (16) is proved.

# F   Proof of Theorem 1

We first prove Proposition 6 and its Corollary in the following Section F.1; subsequently, applying the Corollary proves Theorem 1.

## F.1   Preliminaries

In this section, we first formally define Taylor expansion of the multivariate function in the following Definition 8; Taylor expansion can approximate the function as shown in the following Proposition 5. Subsequently, we consider integrals of functions over a ball, in Proposition 6 and Corollary 2, for proving Theorem 1 in Section F.2.

**Definition 8** (Taylor expansion). Let $d \in \mathbb{N}$ and $q \in \mathbb{N} \cup \{0\}$. For $q$-times differentiable function $f : \mathcal{X} \to \mathbb{R}$, the Taylor polynomial of degree $q \in \mathbb{N} \cup \{0\}$ at point $X_* = (x_{*1}, x_{*2}, \ldots, x_{*d}) \in \mathcal{X}$ is defined as

$$
\mathcal{T}_{q,X_*}[f](X) := \sum_{s=0}^{q} \sum_{|\boldsymbol{i}|=s} \frac{(X - X_*)^{\boldsymbol{i}}}{\boldsymbol{i}!} D^{\boldsymbol{i}} f(X_*),
$$

where $\boldsymbol{i} = (i_1, i_2, \ldots, i_d) \in (\mathbb{N} \cup \{0\})^d$ represents multi-index, $|\boldsymbol{i}| = i_1 + i_2 + \cdots + i_d$, $X^{\boldsymbol{i}} = x_1^{i_1} x_2^{i_2} \cdots x_d^{i_d}$, $\boldsymbol{i}! = i_1! i_2! \cdots i_d!$ and $D^{\boldsymbol{i}} = \frac{\partial^{|\boldsymbol{i}|}}{\partial x_1^{i_1} \partial x_2^{i_2} \cdots \partial x_d^{i_d}}$.

---

**Proposition 5.** Let $d \in \mathbb{N}, \beta > 0$. If $f : \mathcal{X} \to \mathbb{R}$ is $\beta$-Hölder, there exists a function $\varepsilon_{\beta,X_*} : \mathcal{X} \to \mathbb{R}$ such that

$$
f(X) = \mathcal{T}_{\lfloor \beta \rfloor, X_*}[f](X) + \varepsilon_{\beta,X_*}(X),
$$

and $|\varepsilon_{\beta,X_*}(X)| \leq L_\beta \|X - X_*\|_2^\beta \left( \leq L_\beta r^\beta,\ \forall X \in B(X_*; r) \right)$, where $L_\beta$ is a constant for $\beta$-Hölder condition described in Definition 2.

---

*Proof of Proposition 5.* This Proposition 5 immediately follows from the definition of $\beta$-Hölder condition (Definition 2). □

**Proposition 6.** Let $d \in \mathbb{N}, \beta > 0$ and let $f : \mathcal{X} \to \mathbb{R}$ be a $\beta$-Hölder function. Then, for any query $X_* \in \mathcal{X}$, there exists $\tilde{\varepsilon}_\beta \in \mathbb{R}$ such that

$$\int_{B(X_*;r)} f(X)\mathrm{d}X = \sum_{\substack{\boldsymbol{u} \in (\mathbb{N} \cup \{0\})^d \\ |\boldsymbol{u}| \leq \lfloor \beta/2 \rfloor}} \frac{D^{2\boldsymbol{u}} f(X_*)}{(2\boldsymbol{u})!} \frac{g(\boldsymbol{u})}{2|\boldsymbol{u}| + d} r^{2|\boldsymbol{u}| + d} + \tilde{\varepsilon}_\beta, \quad |\tilde{\varepsilon}_\beta| \leq L_\beta r^{\beta + d} \int_{B(\boldsymbol{0};1)} \mathrm{d}x$$

where $g(\boldsymbol{u}) := \frac{2\Gamma(u_1+1/2)\Gamma(u_2+1/2)\cdots\Gamma(u_d+1/2)}{\Gamma(u_1+u_2+\cdots+u_d+d/2)}$ and $\Gamma(u)$ is Gamma function.

*Proof of Propotision 6.* Let $q := \lfloor \beta \rfloor$. In this proof, we first calculate the Taylor expansion $\mathcal{T}_{q,X_*}[\eta](X)$. Then we integrate it over the ball $B(X_*;r)$, by referring to Folland (2001),

Proposition 5 indicates that, there exists a function $\varepsilon_{\beta,X_*}(X)$ such that

$$f(X) = \mathcal{T}_{q,X_*}[f](X) + \varepsilon_{q,X_*}(X) = \sum_{s=0}^{q} \sum_{|\boldsymbol{i}|=s} \frac{(X - X_*)^{\boldsymbol{i}}}{\boldsymbol{i}!} D^{\boldsymbol{i}} f(X_*) + \varepsilon_{\beta,X_*}(X)$$

and $|\varepsilon_{\beta,X_*}(X)| \leq L_\beta r^\beta$, for all $X \in B(X_*;r)$. Therefore, we have

$$\int_{B(X_*;r)} f(X)\mathrm{d}X = \sum_{s=0}^{q} \sum_{|\boldsymbol{i}|=s} \frac{D^{\boldsymbol{i}} f(X_*)}{\boldsymbol{i}!} \underbrace{\int_{B(X_*;r)} (X - X_*)^{\boldsymbol{i}}\mathrm{d}X}_{(\star)} + \underbrace{\int_{B(X_*;r)} \varepsilon_{\beta,X_*}(X)\mathrm{d}X}_{=:\tilde{\varepsilon}_\beta} .$$

We first evaluate the term $(\star)$ in the following.

(a) If at least one entry of $\boldsymbol{i} = (i_1, i_2, \ldots, i_d)$ is odd number, i.e., there exists $j \in [d], u \in \mathbb{N} \cup \{0\}$ such that $i_j = 2u + 1$, it holds that

$$(\star 1) = \int_{B(X_*;r)} (X - X_*)^{\boldsymbol{i}}\mathrm{d}X = \int_{B(\boldsymbol{0};r)} X^{\boldsymbol{i}}\mathrm{d}X = \int_{\substack{B(\boldsymbol{0};r') \\ r' \in [-r,r]}} X_{-j}^{\boldsymbol{i}_{-j}} \underbrace{\left\{ \int_{-\sqrt{r^2 - r'^2}}^{\sqrt{r^2 - r'^2}} x_j^{i_j}\mathrm{d}x_j \right\}}_{=0} \mathrm{d}x_{-j} = 0,$$

where $X_{-j} := (x_1, \ldots, x_{(j-1)}, x_{(j+1)}, \ldots, x_d) \in \mathbb{R}^{d-1}, \boldsymbol{i}_{-j} = (i_1, \ldots, i_{(j-1)}, i_{(j+1)}, \ldots, i_d) \in (\mathbb{N} \cup \{0\})^{d-1}$.

(b) Therefore, in the remaining, we consider the case that all of entries in $\boldsymbol{i} = (i_1, i_2, \ldots, i_d)$ are even numbers, i.e., there exist $u_j \in \mathbb{N} \cup \{0\}$ such that $i_j = 2u_j$ for all $j \in [d]$. It holds that

$$(\star 1) = \int_{B(X_*;r)} (X - X_*)^{\boldsymbol{i}}\mathrm{d}X = \int_{B(\boldsymbol{0};r)} X^{\boldsymbol{i}}\mathrm{d}X$$

$$= \int_0^r \tilde{r}^{|\boldsymbol{i}|+d-1} \underbrace{\int_{\partial B(\boldsymbol{0};\tilde{r})} \tilde{X}^{\boldsymbol{i}}\mathrm{d}\sigma(\tilde{X})}_{=g(\boldsymbol{u}) \ (\because \text{Folland (2001)})} \mathrm{d}\tilde{r}, \quad (\because \text{ polar coordinate})$$

$$= g(\boldsymbol{u}) \int_0^r \tilde{r}^{|\boldsymbol{i}|+d-1}\mathrm{d}\tilde{r} = \frac{1}{|\boldsymbol{i}| + d} g(\boldsymbol{u})$$

where $\partial B(X;\tilde{r})$ denotes a surface of the ball $B(X;\tilde{r})$, $\sigma$ represents $(d-1)$-dimensional surface measure, $g(\boldsymbol{u}) := \frac{2\Gamma(u_1+1/2)\Gamma(u_2+1/2)\cdots\Gamma(u_d+1/2)}{\Gamma(u_1+u_2+\cdots+u_d+d/2)}$ and $\Gamma(u)$ is Gamma function.

Considering above (a) and (b), we have

$$\int_{B(X_*;r)} f(X)\mathrm{d}X = \sum_{\substack{|\boldsymbol{i}| \leq q \\ \boldsymbol{i} = 2\boldsymbol{u}, \boldsymbol{u} \in (\mathbb{N} \cup \{0\})^d}} \frac{D^{\boldsymbol{i}} f(X_*)}{\boldsymbol{i}!} \frac{g(\boldsymbol{u})}{|\boldsymbol{i}| + d} r^{|\boldsymbol{i}| + d} + \tilde{\varepsilon}_\beta$$

$$= \sum_{|\boldsymbol{u}| \leq \lfloor \beta/2 \rfloor} \frac{D^{2\boldsymbol{u}} f(X_*)}{(2\boldsymbol{u})!} \frac{g(\boldsymbol{u})}{2|\boldsymbol{u}| + d} r^{2|\boldsymbol{u}| + d} + \tilde{\varepsilon}_\beta,$$

where $\tilde{\varepsilon}_\beta$ is evaluated by leveraging Proposition 5, i.e.,

$$|\tilde{\varepsilon}_\beta| \leq \int_{B(X_*;r)} |\varepsilon_{\beta,X_*}(X)|\mathrm{d}X \leq \sup_{X \in B(X_*;r)} |\varepsilon_{\beta,X_*}(X)| \int_{B(X_*;r)} \mathrm{d}x \leq L_\beta r^\beta \int_{B(\boldsymbol{0};r)} \mathrm{d}x.$$

Therefore, the assertion is proved. $\qquad\square$

**Corollary 2.** Symbols and assumptions are the same as those of Proposition 6. Then, there exists $\tilde{\varepsilon}_\beta \in \mathbb{R}$ such that

$$\int_{B(X_*;r)} f(X)\mathrm{d}X = \frac{g(\mathbf{0})r^d}{d}f(X_*) + \sum_{c=1}^{\lfloor \beta/2 \rfloor} b_c r^{2c+d} + \tilde{\varepsilon}_\beta, \quad |\tilde{\varepsilon}_\beta| \leq L_\beta r^{\beta+d}\int_{B(\mathbf{0};1)} \mathrm{d}x,$$

where $b_c = b_c(f, X_*) := \frac{1}{2c+d}\sum_{|\mathbf{u}|=c}\frac{D^{2\mathbf{u}}f(X_*)}{(2\mathbf{u})!}g(\mathbf{u})$.

*Proof of Corollary 2.* Proposition 6 immediately proves the assertion. □

### F.2 Main body of the proof

For the function

$$\eta^{(\infty)}(B(X_*;r)) = \frac{\int_{B(X_*;r)}\eta(x)\mu(x)\mathrm{d}x}{\int_{B(X_*;r)}\mu(x)\mathrm{d}x}, \tag{28}$$

Corollary 2 indicates that there exist

$$a_1 = b_1(\eta\mu, X_*), a_2 = b_2(\eta\mu, X_*), \ldots, a_{\lfloor\beta/2\rfloor} = b_{\lfloor\beta/2\rfloor}(\eta\mu, X_*) \in \mathbb{R},$$
$$b_1 = b_1(\mu, X_*), b_2 = b_2(\mu, X_*), \ldots, b_{\lfloor\beta/2\rfloor} = b_{\lfloor\beta/2\rfloor}(\mu, X_*) \in \mathbb{R} \tag{29}$$

and $\tilde{\varepsilon}_\beta^{(1)}, \tilde{\varepsilon}_\beta^{(2)} \in \mathbb{R}$ such that

$$(28) = \frac{\frac{g(\mathbf{0})r^d}{d}\eta(X_*)\mu(X_*) + \sum_{c=1}^{\lfloor\beta/2\rfloor}a_c r^{2c+d} + \tilde{\varepsilon}_\beta^{(1)}}{\frac{g(\mathbf{0})r^d}{d}\mu(X_*) + \sum_{c=1}^{\lfloor\beta/2\rfloor}b_c r^{2c+d} + \tilde{\varepsilon}_\beta^{(2)}}, \qquad |\tilde{\varepsilon}_\beta^{(1)}|, |\tilde{\varepsilon}_\beta^{(2)}| \leq L_\beta r^{\beta+d}\int_{B(\mathbf{0};1)}\mathrm{d}x, \tag{30}$$

since $\mu$ and $\eta\mu$ are $\beta$-Hölder. Both the numerator and denominator are divided by $r^d$, then for sufficiently small $r > 0$, the asymptotic expansion is of the form

$$(30) = \eta(X_*) + \sum_{c=1}^{\lfloor\beta/2\rfloor}b_c^*(X_*)r^{2c} + \delta_{\beta,r}(X_*), \tag{31}$$

where $\delta_{\beta,r}(X_*) = O(r^{2\lfloor\beta/2\rfloor+2}) + O(r^\beta)$. The two error terms are in fact combined as $\delta_{\beta,r}(X_*) = O(r^\beta)$, because $2\lfloor\beta/2\rfloor + 2 \geq \beta$. Thus, by specifying a sufficiently small $\tilde{r} > 0$, the error term is bounded as $\delta_{\beta,r}(X_*) < L_\beta^*(X_*)r^\beta$ for $r \in (0, \tilde{r}]$ with a continuous function $L_\beta^*(X_*)$. For $L_\beta^* = \sup_{X\in\mathcal{S}(\mu)}L_\beta^*(X_*) < \infty$, we have

$$(31) = \eta(X_*) + \sum_{c=1}^{\lfloor\beta/2\rfloor}b_c^*(X_*)r^{2c} + \delta_{\beta,r}(X_*), \quad |\delta_{\beta,r}(X_*)| < L_\beta^* r^\beta, \quad (\forall r \in (0,\tilde{r}], X_* \in \mathcal{S}(\mu)).$$

Thus proving the assertion. Note that, by rearranging the terms of order $r^{2+d}$, we obtain the equation

$$\frac{g(\mathbf{0})}{d}\mu(X_*)b_1^* + \eta(X_*)b_1 = a_1,$$

where $a_1 := \frac{1}{2+d}\sum_{|\mathbf{u}|=1}\frac{D^{2\mathbf{u}}(\eta(X_*)\mu(X_*))}{(2\mathbf{u})!}g(\mathbf{u}), b_1 := \frac{1}{2+d}\sum_{|\mathbf{u}|=1}\frac{D^{2\mathbf{u}}\mu(X_*)}{(2\mathbf{u})!}g(\mathbf{u})$; subsequently, solving the equation yields

$$b_1^* = \frac{d}{2+d}\frac{1}{\mu(X_*)}\sum_{|\mathbf{u}|=1}\left\{\frac{D^{2\mathbf{u}}(\eta(X_*)\mu(X_*))}{(2\mathbf{u})!} - \frac{\eta(X_*)D^{2\mathbf{u}}\mu(X_*)}{(2\mathbf{u})!}\right\}\frac{g(\mathbf{u})}{g(\mathbf{0})}$$

$$= \frac{d}{2+d}\frac{1}{\mu(X_*)}\frac{1}{2}\{\Delta\eta(X_*)\mu(X_*) + \eta(X_*)\Delta\mu(X_*)\}\underbrace{\frac{2\Gamma(1/2)^{d-1}\Gamma(3/2)/\Gamma(1+d/2)}{2\Gamma(1/2)^d/\Gamma(d/2)}}_{=\frac{\Gamma(3/2)/\Gamma(1+d/2)}{\Gamma(1/2)/\Gamma(d/2)}=\frac{1/2}{d/2}=\frac{1}{d}}$$

$$= \frac{1}{2d+4}\frac{1}{\mu(X_*)}\{\Delta[\eta(X_*)\mu(X_*)] + \eta(X_*)\Delta\mu(X_*)\}.$$

In general, $b_1^* \neq 0$, thus $\gamma = 2$ for $\beta > 2$. For the case of $\beta = 2$, we have $\lfloor\beta/2\rfloor = 0$, thus $(30) = \eta(X_*) + O(r^\beta)$, meaning $\gamma = 2$. □

# G  Proof of Theorem 2

We basically follow the proof of Chaudhuri & Dasgupta (2014) Theorem 4(b). In Section G.1, we first define symbols used in this proof. In Section G.2, we describe the sketch of the proof and main differences between our proof and that of Chaudhuri & Dasgupta (2014) 4(b). Section G.3 shows the main body of the Proof, by utilizing several Lemmas listed in Section G.4.

## G.1  Definitions of symbols

- **$k$ and radius $r$:** We first specify a real-valued vector $\boldsymbol{\ell} = (\ell_1, \ell_2, \ldots, \ell_V)^\top \in \mathbb{R}^V$ satisfying $\ell_1 = 1 < \ell_2 < \cdots < \ell_V$. $k_{1,n} \asymp n^{-2\beta/(2\beta+d)}$ is assumed in (C-1), and in (C-2), $\{k_{v,n}\}$ are specified so that

$$k_{v,n} = \min\{k \in [n] \mid \|X_{(k)} - X_*\|_2 \geq \ell_v r_{1,n}\}, \quad \forall v \in \{2, 3, \ldots, V\}$$

  from $r_{1,n} := \|X_{(k_{1,n})} - X_*\|_2$. Then, for $r_{v,n} := \|X_{(k_{v,n})} - X_*\|_2$, $v = 2, \ldots, V$, we have $r_{v,n}/r_{1,n} \to \ell_v$.

- **Estimators:** Similarly to Supplement E, we denote the $k$-NN estimators and MS-$k$-NN estimator by where

$$
\begin{array}{ll}
\text{(Finite $k$-NN)} & \varphi_{n,k} = \varphi_{n,k}(X_*) := \frac{1}{k}\sum_{i=1}^k Y_{(i;X_*)} \in \mathbb{R} \\
\text{(Finite $k$-NN vector)} & \boldsymbol{\varphi}_{n,\boldsymbol{k}} = \boldsymbol{\varphi}_{n,\boldsymbol{k}}(X_*) := (\varphi_{n,k_1}(X_*), \varphi_{n,k_2}(X_*), \ldots, \varphi_{n,k_V}(X_*))^\top \in \mathbb{R}^V, \\
\text{(Finite MS-$k$-NN)} & \rho_{n,\boldsymbol{k}} = \rho_{n,\boldsymbol{k}}(X_*) := \boldsymbol{z}_{n,\boldsymbol{k}}(X_*)^\top \boldsymbol{\varphi}_{n,\boldsymbol{k}}(X_*) \in \mathbb{R},
\end{array}
$$

$\boldsymbol{z}_{n,\boldsymbol{k}}(X_*) \in \mathbb{R}^V$ denotes vectror $\boldsymbol{z}$ considered in Supplement E, i.e.,

$$\boldsymbol{z}_{n,\boldsymbol{k}}(X_*) := \frac{(\boldsymbol{I} - \mathcal{P}_{\boldsymbol{R}_{n,\boldsymbol{k}}(X_*)})\boldsymbol{1}}{V - \boldsymbol{1}^\top \mathcal{P}_{\boldsymbol{R}_{n,\boldsymbol{k}}(X_*)}\boldsymbol{1}}$$

  where $\mathcal{P}_{\boldsymbol{R}} := \boldsymbol{R}(\boldsymbol{R}^\top \boldsymbol{R})^{-1}\boldsymbol{R}^\top$ and the $(i,j)$-th entry of the matrix $\boldsymbol{R}_{n,\boldsymbol{k}}(X_*)$ is $r_{k_i}^{2j} = \|X_{(k_i)} - X_*\|_2^{2j}$. Whereas the vector $\boldsymbol{z}_{n,k}(X_*)$ is simply denoted by $\boldsymbol{z}$ in the above discussion, here we emphasize the dependence to the sample $\mathcal{D}_n$, parameters $\boldsymbol{k} = (k_1, k_2, \ldots, k_V)$ and the query $X_*$.

  We here define the asymptotic variants of the estimators by where $\boldsymbol{r} = (r_1, r_2, \ldots, r_V)$,

$$
\begin{array}{ll}
\text{(Asymptotic $k$-NN)} & \varphi_r^{(\infty)} = \varphi_r^{(\infty)}(X_*) := \eta^{(\infty)}(B(X_*; r)) \in \mathbb{R}, \\
\text{(Asymptotic $k$-NN vector)} & \boldsymbol{\varphi}_{\boldsymbol{r}}^{(\infty)} = \boldsymbol{\varphi}_{\boldsymbol{r}}^{(\infty)}(X_*) := (\varphi_{r_1}^{(\infty)}(X_*), \varphi_{r_2}^{(\infty)}(X_*), \ldots, \varphi_{r_V}^{(\infty)}(X_*)) \in \mathbb{R}^V, \\
\text{(Asymptotic MS-$k$-NN)} & \rho_{\boldsymbol{r}}^{(\infty)} = \rho_{\boldsymbol{r}}^{(\infty)}(X_*) := \boldsymbol{z}_{\boldsymbol{r}}^\top \boldsymbol{\varphi}_{\boldsymbol{r}}^{(\infty)}(X_*) \in \mathbb{R},
\end{array}
$$

$$\boldsymbol{z}_{\boldsymbol{r}} = \frac{(\boldsymbol{I} - \mathcal{P}_{\boldsymbol{R}})\boldsymbol{1}}{V - \boldsymbol{1}^\top \mathcal{P}_{\boldsymbol{R}}\boldsymbol{1}},$$

  and the $(i,j)$-th entry of the matrix $\boldsymbol{R}$ is $r_i^{2j}$.

- **Point-wise errors** for $(X_*, Y_*) \in \mathcal{X} \times \{0, 1\}$ are defined as

$$R_{n,\boldsymbol{k}}(X_*, Y_*) := \mathbb{1}(\underbrace{\rho_{n,\boldsymbol{k}}(X_*)}_{\text{Finite MS-$k$-NN}} \neq Y_*), \quad R_*(X_*, Y_*) := \mathbb{1}(g_*(X_*) \neq Y_*),$$

  where $g_*(X) := \mathbb{1}(\eta(X) \geq 1/2)$ is the Bayes-optimal classifier equipped with $\eta(X) := \mathbb{E}(Y \mid X)$.

- **A minimum radius** whose measure of the ball is larger than $t > 0$, i.e.,

$$\tilde{r}_t(X) := \inf\left\{r > 0 \,\middle|\, \int_{B(X_*; r)} \mu(X)\mathrm{d}X \geq t\right\}.$$

- **Sets for the decision boundary with margins** are defined as

$$
\begin{aligned}
\mathcal{X}_{t,\Delta}^+ &:= \left\{X \in \mathcal{S}(\mu) \mid \eta(X) > \frac{1}{2}, \quad \rho_{r\boldsymbol{\ell}}^{(\infty)}(X) \geq \frac{1}{2} + \Delta, \quad \forall r \leq \tilde{r}_t(X)\right\}, \\
\mathcal{X}_{t,\Delta}^- &:= \left\{X \in \mathcal{S}(\mu) \mid \eta(X) < \frac{1}{2}, \quad \rho_{r\boldsymbol{\ell}}^{(\infty)}(X) \leq \frac{1}{2} - \Delta, \quad \forall r \leq \tilde{r}_t(X)\right\}, \\
\partial_{t,\Delta} &:= \mathcal{X} \setminus (\mathcal{X}_{t,\Delta}^+ \cup \mathcal{X}_{t,\Delta}^-),
\end{aligned}
$$

  where $\mathcal{S}(\mu)$ is defined in (5), and $r$ is meant for $r_1$.

## G.2 Sketch of the proof

**Sketch of the proof:** We mainly follow the proof of Chaudhuri & Dasgupta (2014) Theorem 4(b), that proves the convergence rate for the unweighted $k$-NN estimator. Similarly to Chaudhuri & Dasgupta (2014) Lemma 7, we first consider decomposing the difference between point-wise errors $R_{n,\boldsymbol{k}}(X_*, Y_*) - R_*(X_*, Y_*)$ as shown in the following Lemma 1; this Lemma plays an essential role for proving Theorem 1.

Subsequently, we consider the following two steps using Lemma 1–7:

(i) taking expectation of the decomposition w.r.t. sample $\mathcal{D}_n$ for showing point-wise excess risk,
    (cf. Chaudhuri & Dasgupta (2014) Lemma 20)

(ii) further taking expectation w.r.t. the query $(X_*, Y_*)$, and evaluate the convergence rate.
    (cf. Chaudhuri & Dasgupta (2014) Lemma 21)

Then, the assertion is proved.

**Main difference** between the Proof of Chaudhuri & Dasgupta (2014) and ours is bias evaluation. Chaudhuri & Dasgupta (2014) leverages the $\gamma$-neighbour average smoothness condition

$$\text{(asymptotic bias of } k\text{-NN)} \qquad | \underbrace{\varphi_r^{(\infty)}(X_*)}_{\text{(asymptotic) } k\text{-NN}} -\eta(X_*)| \le L_\gamma r^\gamma,$$

that represents the asymptotic bias of the $k$-NN, where $\gamma$ is upper-bounded by 2 even if highly-smooth function is employed ($\beta \gg 2$; Theorem 1). However, MS-$k$-NN asymptotically satisfies an inequality

$$\text{(asymptotic bias of MS-}k\text{-NN)} \qquad | \underbrace{\rho_{r\boldsymbol{\ell}}^{(\infty)}(X_*)}_{\text{(asymptotic) MS-}k\text{-NN}} -\eta(X_*)| \le L_\beta^{**} r^\beta$$

for any $\beta > 0$, as formally described in Lemma 2. By virtue of the smaller asymptotic bias, Lemma 4 proves that smaller margin is required for the decision boundary, in order to evaluate the convergence rate; it results in the faster convergence rate.

Although the the bias evaluation is different, variance evaluation for MS-$k$-NN is consequently almost similar to the $k$-NN, as MS-$k$-NN can be regarded as a linear combination of several $k$-NN estimators, i.e.,

$$\underbrace{\rho_{n,\boldsymbol{k}}(X_*)}_{\text{MS-}k\text{-NN estimator}} = \boldsymbol{z}_{n,\boldsymbol{k}}(X_*)^\top \underbrace{\boldsymbol{\varphi}_{n,\boldsymbol{k}}(X_*)}_{k\text{-NN estimators}};$$

we adapt several Lemmas in Chaudhuri & Dasgupta (2014) to our setting, for proving our Theorem 2.

## G.3 Main body of the proof

See the following Section G.4 for Lemma 1–7 used in this proof. Throughout this proof, we assume that $X_* \in \mathcal{S}(\mu)$, as Cover & Hart (1967) proves that $\mathbb{P}(X_* \in \mathcal{S}(\mu)) = 1$; the remaining $X_* \notin \mathcal{S}(\mu)$ can be ignored.

Let $n \in \mathbb{N}, k_{1,n} \asymp n^{2\beta/(2\beta+d)}, t_n := 2k_{1,n}/n, \Delta_o := L_\beta^{***} t^{\beta/d}$ where $L_\beta^{***} \in (0, \infty)$ is a constant defined in Lemma 4, and let $\Delta(X) := |\eta(X) - 1/2|$ denotes the difference between the underlying conditional expectation $\eta(X)$ from the decision boundary $1/2$.

By specifying arbitrary $i_o \in \mathbb{N}$ and $\Delta_{i_o} := 2^{i_o}\Delta_o$, we consider the following two steps (i) and (ii) for proving Theorem 2. In step (i), queries are first classified into two different cases, i.e., $\Delta(X_*) \le \Delta_{i_o}$ and $\Delta(X_*) > \Delta_{i_o}$. Thus $i_o$ regulates the margin near the decision boundary, and it will be specified as $i_o = \max\{1, \lceil \log_2 \sqrt{\frac{2(\alpha+2)}{k_{1,n}\Delta_o^2}} \rceil\}$. For each case, we take expectation of the difference between point-wise errors $R_{n,\boldsymbol{k}}(X_*, Y_*) - R_*(X_*, Y_*)$ with respect to the sample $\mathcal{D}_n$. Subsequently, (ii) we further take its expectation with respect to the query $(X_*, Y_*)$; the assertion is then proved. Note that these steps (i) and (ii) correspond to Chaudhuri & Dasgupta (2014) Lemma 20 and 21, respectively.

(i) We first consider the case $\Delta(X_*) \le \Delta_{i_o}$. Then, we have

$$\mathbb{E}_{\mathcal{D}_n}(R_{n,\boldsymbol{k}}(X_*, Y_*) - R_*(X_*, Y_*)) \le |1 - 2\eta(X_*)|\mathbb{E}_{\mathcal{D}_n}\{\mathbb{1}(\rho_{n,\boldsymbol{k}}(X_*) \ne g_*(X_*))\}$$
$$(\because \text{Devroye et al. (1996) Theorem 2.2})$$
$$\le |1 - 2\eta(X_*)|$$
$$\le 2\Delta(X_*)$$
$$\le 2\Delta_{i_o}. \tag{32}$$

We second consider the case $\Delta(X_*) > \Delta_{i_o}$. Assuming that $\eta(X_*) > 1/2$ without loss of generality, it holds for $r = r_{1,n} := \|X_{(k_{1,n})} - X_*\|_2$ that

$$\mathbb{E}_{\mathcal{D}_n}\left\{R_{n,\boldsymbol{k}}(X_*, Y_*) - R_*(X_*, Y_*)\right\}$$

$$\leq |1 - 2\eta(X_*)|\mathbb{E}_{\mathcal{D}_n}\left\{\mathbb{1}(\rho_{n,\boldsymbol{k}}(X_*) \neq g_*(X_*))\right\}$$

$$(\because \text{Devroye et al. (1996) Theorem 2.2})$$

$$\leq 2\Delta(X_*)\mathbb{E}_{\mathcal{D}_n}\left\{\mathbb{1}(\rho_{n,\boldsymbol{k}}(X_*) \neq g_*(X_*))\right\}$$

$$(\because |1 - 2\eta(X_*)| \leq 2\Delta(X_*))$$

$$\leq 2\Delta(X_*)\mathbb{E}_{\mathcal{D}_n}\bigg\{\mathbb{1}(X_* \in \partial_{t_n, \Delta(X_*) - \Delta_{i_o}})$$

$$+ \mathbb{1}\left(|\rho_{n,\boldsymbol{k}}(X_*) - \rho_{r\boldsymbol{\ell}}^{(\infty)}(X_*)| \geq \frac{\Delta(X_*) - \Delta_{i_o}}{2}\right)$$

$$+ \mathbb{1}\left(|\rho_{n,\boldsymbol{k}}(X_*) - \eta(X_*)| \geq \frac{\Delta(X_*) - \Delta_{i_o}}{2}\right)$$

$$+ \mathbb{1}(\|X_{(k_{1,n})} - X_*\|_2 > \tilde{r}_{t_n}(X_*))\bigg\}$$

$$(\because \text{Lemma 1 with } \Delta := \Delta(X_*) - \Delta_{i_o} \in [0, 1/2])$$

$$\leq 2\Delta(X_*)\mathbb{E}_{\mathcal{D}_n}\bigg\{\mathbb{1}\left(|\rho_{n,\boldsymbol{k}}(X_*) - \rho_{r\boldsymbol{\ell}}^{(\infty)}(X_*)| \geq \frac{\Delta(X_*) - \Delta_{i_o}}{2}\right)$$

$$+ \mathbb{1}\left(|\rho_{n,\boldsymbol{k}}(X_*) - \eta(X_*)| \geq \frac{\Delta(X_*) - \Delta_{i_o}}{2}\right)$$

$$+ \mathbb{1}(\|X_{(k_{1,n})} - X_*\|_2 > \tilde{r}_{t_n}(X_*))\bigg\} \qquad (\because \text{Lemma 4, i.e., } X_* \notin \partial_{t_n, \Delta(X_*) - \Delta_{i_o}})$$

$$\leq 2\Delta(X_*)\bigg\{\mathbb{P}_{\mathcal{D}_n}\left(|\rho_{n,\boldsymbol{k}}(X_*) - \rho_{r\boldsymbol{\ell}}^{(\infty)}(X_*)| \geq \frac{\Delta(X_*) - \Delta_{i_o}}{2}\right)$$

$$+ \mathbb{P}_{\mathcal{D}_n}\left(|\rho_{n,\boldsymbol{k}}(X_*) - \eta(X_*)| \geq \frac{\Delta(X_*) - \Delta_{i_o}}{2}\right)$$

$$+ \mathbb{P}_{\mathcal{D}_n}(\|X_{(k_{1,n})} - X_*\|_2 > \tilde{r}_{t_n}(X_*))\bigg\} \quad (\because \mathbb{E}_{\mathcal{D}_n}(\mathbb{1}(\mathcal{A})) = \mathbb{P}_{\mathcal{D}_n}(\mathcal{A}) \text{ for any event } \mathcal{A})$$

$$\lesssim \Delta(X_*)\bigg\{\exp\left(-C_1 k_{1,n}(\Delta(X_*) - \Delta_{i_o})^2\right) + \exp\left(-C_2 k_{1,n}(\Delta(X_*) - \Delta_{i_o})^2\right)$$

$$+ \exp(-L_{\boldsymbol{\ell}} n^{\beta/(\beta+d)}(\Delta(X_*) - \Delta_{i_o})) + \exp(-3k_{1,n}/2)(1 + o(1))$$

$$+ \exp(-n) + \exp\left(-\frac{k_{1,n}}{2}\left(1 - \frac{k_{1,n}}{nt_n}\right)^2\right)\bigg\} \quad (\because \text{Lemma 5, 6 and 7 with } \delta = k_{1,n}/nt)$$

$$\lesssim \Delta(X_*)\exp\left(-C_2 k_{1,n}(\Delta(X_*) - \Delta_{i_o})^2\right) + + \exp(-3k_{1,n}/2)(1 + o(1)) + \exp(-k_{1,n}/8) \qquad (33)$$

$$\left(\because t_n = 2(k_{1,n}/n) \text{ indicates that } \frac{k_{1,n}}{2}\left(1 - \frac{k_{1,n}}{nt_n}\right)^2 = \frac{k_{1,n}}{2}\left(1 - \frac{1}{2}\right)^2 = \frac{k_{1,n}}{8}\right),$$

$$\lesssim \Delta(X_*)\exp\left(-C_2 k_{1,n}(\Delta(X_*) - \Delta_{i_o})^2\right) + \exp(-3k_{1,n}/2)(1 + o(1)),$$

where $C_2 = C_1/2 = 1/16V^2 L_{\boldsymbol{z}}^2$ is defined in Lemma 6.

(ii) Excess risk of the misclassification error rate is then evaluated by

$$\varepsilon(\rho_{n,\boldsymbol{k}}) = \mathbb{E}_{X_*, Y_*}\left\{\mathbb{E}_{\mathcal{D}_n}(R_{n,\boldsymbol{k}}(X_*, Y_*) - R_*(X_*, Y_*))\right\}$$

$$= \underbrace{\mathbb{P}_{X_*, Y_*}(\Delta(X_*) \leq \Delta_{i_o})}_{\leq L_\alpha \Delta_{i_o}^\alpha \ (\because \alpha\text{-margin cond.})} \mathbb{E}_{X_*, Y_*}(\underbrace{\mathbb{E}_{\mathcal{D}_n}\left\{R_{n,\boldsymbol{k}}(X_*, Y_*) - R_*(X_*, Y_*)\right\}}_{\leq 2\Delta_{i_o} \ (\because \text{ineq. (32)})} \mid \Delta(X_*) \leq \Delta_{i_o})$$

$$+ \underbrace{\mathbb{P}_{X_*, Y_*}(\Delta(X_*) > \Delta_{i_o})}_{\leq 1} E_{X_*, Y_*}(\mathbb{E}_{\mathcal{D}_n}\left\{R_{n,\boldsymbol{k}}(X_*, Y_*) - R_*(X_*, Y_*)\right\} \mid \Delta(X_*) > \Delta_{i_o})$$

$$\lesssim \Delta_{i_o}^{1+\alpha} + E_{X_*, Y_*}(\underbrace{\mathbb{E}_{\mathcal{D}_n}\left\{R_{n,\boldsymbol{k}}(X_*, Y_*) - R_*(X_*, Y_*)\right\}}_{\text{(evaluated by ineq. (33))}} \mid \Delta(X_*) > \Delta_{i_o})$$

$$\lesssim \Delta_{i_o}^{1+\alpha} + \underbrace{\mathbb{E}_{X_*, Y_*}\left(\Delta(X_*)\exp(-C_2 k_{1,n}(\Delta(X_*) - \Delta_o)^2)\mathbb{1}(\Delta(X_*) > \Delta_{i_o})\right)}_{\lesssim \Delta_{i_o}^{1+\alpha} \quad (\because \text{similarly to Proof of Lemma 20 in Chaudhuri \& Dasgupta (2014))}} + \exp(-3k_{1,n}/2)(1 + o(1))$$

$$\lesssim \Delta_{i_o}^{1+\alpha} + \exp(-3k_{1,n}/2)(1 + o(1)).$$

If we set $i_0 = \max\{1, \lceil \log_2 \sqrt{\frac{2(\alpha+2)}{k_{1,n}\Delta_o^2}} \rceil\}$,

$$
\begin{aligned}
\varepsilon(\hat{\eta}_{n,\boldsymbol{k}}^{(\text{MS-}k\text{NN})}) &= \varepsilon(\rho_{n,\boldsymbol{k}}) \\
&\lesssim \Delta_o^{1+\alpha} + \exp(-3k_{1,n}/2)(1 + o(1)) \\
&\lesssim (2^{i_o})^{1+\alpha}\Delta_o^{1+\alpha} + \exp(-3k_{1,n}/2)(1 + o(1)) \\
&\lesssim \left(\max\left\{1, \sqrt{\frac{2(\alpha+2)}{k_{1,n}\Delta_o^2}}\right\}\right)^{1+\alpha}\Delta_o^{1+\alpha} + \exp(-3k_{1,n}/2)(1 + o(1)) \\
&\lesssim \max\left\{\Delta_o, \sqrt{\frac{1}{k_{1,n}}}\right\}^{1+\alpha} + \exp(-3k_{1,n}/2)(1 + o(1)) \\
&\lesssim \max\left\{t_n^{\beta/d}, \sqrt{\frac{1}{k_{1,n}}}\right\}^{1+\alpha} + \exp(-3k_{1,n}/2)(1 + o(1)) && (\Delta_o \asymp t_n^{\beta/d}) \\
&\lesssim \max\left\{\left(\frac{k_{1,n}}{n}\right)^{\beta/d}, \sqrt{\frac{1}{k_{1,n}}}\right\}^{1+\alpha} + \exp(-3k_{1,n}/2)(1 + o(1)) && (\because t_n \asymp k_{1,n}/n).
\end{aligned}
$$

Recalling that $k_{1,n} \asymp n^{2\beta/(2\beta+d)}$, the assertion is proved as

$$\varepsilon(\hat{\eta}_{n,\boldsymbol{k}}^{(\text{MS-}k\text{NN})}) \lesssim n^{-(1+\alpha)\beta/(2\beta+d)}.$$

$\square$

### G.4 Lemmas

We here list Lemma 1–7 used in the proof for Theorem 2. Roughly speaking,

- Lemma 1 indicates the decomposition of the point-wise error.
  (cf. Chaudhuri & Dasgupta (2014) Lemma 7)

- Lemma 2 indicates the bias evaluation of MS-$k$-NN.

- Lemma 3 indicates the convergence rate of $\|\boldsymbol{z}_{n,\boldsymbol{k}}(X_*) - \boldsymbol{z}_{r\ell}\|_\infty$.

- Lemma 4 adapts the first part of Chaudhuri & Dasgupta (2014) Lemma 20 from unweighted $k$-NN to MS-$k$-NN.

- Lemma 5 and 6 indicate the convergence rates related to the bias and variance evaluation of the MS-$k$-NN.
  (cf. Chaudhuri & Dasgupta (2014) Lemma 9)

- Lemma 7 indicates how fast the radius $r > 0$ decreases to 0 as $k$ increases.
  (cf. Chaudhuri & Dasgupta (2014) Lemma 8)

Similarly to Chaudhuri & Dasgupta (2014) Lemma 7, we prove the following Lemma 1, that decomposes the point-wise error into four different parts.

---

**Lemma 1.** Let $g_{n,\boldsymbol{k}}$ be the MS-$k$-NN classifier based on sample $\mathcal{D}_n$, and let $X_* \in \mathcal{S}(\mu), t \in [0,1], \Delta \in [0, 1/2]$. Then, it holds for $r = r_{1,n} := \|X_{(k_{1,n})} - X_*\|_2$ that

$$\mathbb{1}(g_{n,\boldsymbol{k}}(X_*) \neq g_*(X_*)) \leq \mathbb{1}(X_* \in \partial_{t,\Delta}) \tag{34}$$

$$+ \mathbb{1}(|\rho_{n,\boldsymbol{k}}(X_*) - \rho_{r\ell}^{(\infty)}(X_*)| \geq \Delta/2) \tag{35}$$

$$\mathbb{1}(|\rho_{n,\boldsymbol{k}}(X_*) - \eta(X_*)| \geq \Delta/2) \tag{36}$$

$$+ \mathbb{1}(r > \tilde{r}_t(X_*)). \tag{37}$$

---

*Proof of Lemma 1.* Let $\mathcal{A}$ be an event that $g_{n,\boldsymbol{k}}(X_*) \neq g_*(X_*)$, and let $\mathcal{B}_1, \mathcal{B}_2, \mathcal{B}_3, \mathcal{B}_4$ be events defined by the indicator functions (34)–(37), respectively. Then, it suffices to prove $\mathcal{A} \Rightarrow [\mathcal{B}_1 \vee \mathcal{B}_2 \vee \mathcal{B}_3 \vee \mathcal{B}_4]$ or its contrapositive $[(\neg\mathcal{B}_1) \wedge (\neg\mathcal{B}_2) \wedge (\neg\mathcal{B}_3) \wedge (\neg\mathcal{B}_4)] \Rightarrow \neg\mathcal{A}$, where $\neg$ represents the negation. Here, we prove the contrapositive.

$\neg\mathcal{B}_1$ indicates that $X_* \in \mathcal{X}_{t,\Delta}^+$ or $X_* \in \cup\mathcal{X}_{t,\Delta}^-$.

- We here consider the former case $X_* \in \mathcal{X}_{t,\Delta}^+$; then, $\neg \mathcal{B}_4$, i.e., $r \leq \tilde{r}_t(X)$, indicates that

$$\rho_{r\boldsymbol{\ell}}^{(\infty)}(X_*) \geq \frac{1}{2} + \Delta (> 1/2). \tag{38}$$

$\neg \mathcal{B}_2$ and $\neg \mathcal{B}_3$ represent

$$|\rho_{n,\boldsymbol{k}} - \rho_{r\boldsymbol{\ell}}^{(\infty)}(X_*)| < \Delta/2, \quad |\rho_{n,\boldsymbol{k}} - \eta(X_*)| < \Delta/2, \tag{39}$$

respectively; above inequalities (38) and (39) indicate

$$\eta(X_*) \geq \rho_{r\boldsymbol{\ell}}^{(\infty)}(X_*) - |\rho_{n,\boldsymbol{k}} - \rho_{r\boldsymbol{\ell}}^{(\infty)}(X_*)| - |\rho_{n,\boldsymbol{k}} - \eta(X_*)| > \frac{1}{2} + \Delta - \Delta/2 - \Delta/2 = 1/2. \tag{40}$$

(38) and (40) prove that both of corresponding classifiers output the same label 1, whereupon $\neg \mathcal{A}$.

- Similarly, for the latter case $X_* \in \mathcal{X}_{t,\Delta}^-$, both classifiers output 0 and thus $\neg \mathcal{A}$.

Therefore, the assertion is proved. $\qquad\square$

---

**Lemma 2.** Assuming the assumption (C-3), i.e., there exists $L_{\boldsymbol{z}} \in (0, \infty)$ such that $\|\boldsymbol{z}_{\boldsymbol{\ell}}\|_\infty < L_{\boldsymbol{z}}$. Then, there exist $\tilde{r}, L_\beta^{**} \in (0, \infty)$ such that

$$|\rho_{r\boldsymbol{\ell}}^{(\infty)}(X_*) - \eta(X_*)| \leq L_\beta^{**} r^\beta, \quad (\forall X_* \in \mathcal{X}, r \in (0, \tilde{r}]).$$

---

*Proof of Lemma 2.* Theorem 1 proves

$$\underbrace{\varphi_r^{(\infty)}(X_*)}_{\text{asymptotic } k\text{-NN}} = \eta(X_*) + \sum_{c=1}^{\lfloor \beta/2 \rfloor} b_c^* r^{2c} + \delta_r(X_*), \quad |\delta_r(X_*)| \leq L_\beta^* r^\beta,$$

for all $X_* \in \mathcal{S}(\mu), r \in (0, \tilde{r}]$, for some $\tilde{r} \in (0, \infty)$; we have a simultaneous equation

$$\boldsymbol{\varphi}_{r\boldsymbol{\ell}}^{(\infty)}(X_*) = \boldsymbol{A}_{r\boldsymbol{\ell}}(X_*)\boldsymbol{b}_*(X_*) + \boldsymbol{\delta}_r(X_*), \quad \|\boldsymbol{\delta}_r(X_*)\|_\infty \leq L_\beta^* r^\beta \quad (\forall X_* \in \mathcal{X}, r \in (0, \tilde{r}]),$$

where $\boldsymbol{A}_{r\boldsymbol{\ell}} = \boldsymbol{A}_{r\boldsymbol{\ell}}(X_*) = (\boldsymbol{1} \ \boldsymbol{R}(X_*)) \in \mathbb{R}^{V \times (C+1)}$ is defined as same as $\boldsymbol{A}$ in (26) with the radius vector $\boldsymbol{r} = (r_1, r_2, \ldots, r_V) = r\boldsymbol{\ell}$, and the entries in $\boldsymbol{b}_*(X_*) = (\eta(X_*), b_1^*, b_2^*, \ldots, b_{\lfloor \beta/2 \rfloor}^*)$ are specified in Theorem 1. Denoting the first entry of the vector $\boldsymbol{b}$ by $[\boldsymbol{b}]_1$,

$$| \underbrace{\rho_{r\boldsymbol{\ell}}^{(\infty)}(X_*)}_{\text{asymptotic MS-}k\text{-NN}} -\eta(X_*)| \leq |[(\boldsymbol{A}^\top \boldsymbol{A})^{-1} \boldsymbol{A}^\top \boldsymbol{\varphi}_{r\boldsymbol{\ell}}^{(\infty)}]_1 - \eta(X_*)|$$

$$\leq \left| [(\boldsymbol{A}^\top \boldsymbol{A})^{-1} \boldsymbol{A}^\top \{\boldsymbol{A}(X_*)\boldsymbol{b}_*(X_*) + \boldsymbol{\delta}_r(X_*)\}]_1 - \eta(X_*) \right|$$

$$= | \underbrace{[\boldsymbol{b}_*(X_*)]_1}_{=\eta(X_*)} + \underbrace{[(\boldsymbol{A}^\top \boldsymbol{A})^{-1} \boldsymbol{A}^\top \boldsymbol{\delta}_r(X_*)]_1}_{=\boldsymbol{z}_{r\boldsymbol{\ell}}^\top \boldsymbol{\delta}_r(X_*)} -\eta(X_*)|$$

$$= |\boldsymbol{z}_{r\boldsymbol{\ell}}^\top \boldsymbol{\delta}_r(X_*)|$$

$$\leq \underbrace{\|\boldsymbol{z}_{r\boldsymbol{\ell}}\|_\infty}_{\leq L_{\boldsymbol{z}}} \underbrace{\|\boldsymbol{\delta}_r(X_*)\|_\infty}_{\leq L_\beta^* r^\beta} \quad (\because \boldsymbol{z}_{r\boldsymbol{\ell}} = \boldsymbol{z}_{\boldsymbol{\ell}}, \ \forall r > 0)$$

$$\leq L_{\boldsymbol{z}} L_\beta^* r^\beta.$$

Specifying $L_\beta^{**} := L_{\boldsymbol{z}} L_\beta^*$ leads to the assertion. $\qquad\square$

---

**Lemma 3.** Assuming that $X_* \in \mathcal{S}(\mu)$, (C-1) $k_{1,n} \asymp n^{2\beta/(2\beta+d)}$ and (C-2) $k_{v,n} = \min\{k \in [n] \mid \|X_{(k)} - X_*\|_2 \geq \ell_v r_{1,n}\}$ where $r = r_{1,n} := \|X_{(k_{1,n})} - X_*\|_2$. Then, for sufficiently large $n \in \mathbb{N}$, there exists $L_{\boldsymbol{\ell}} > 0$ such that

$$\mathbb{P}\left(\|\boldsymbol{z}_{n,\boldsymbol{k}}(X_*) - \boldsymbol{z}_{r\boldsymbol{\ell}}\|_\infty > \Delta\right) \lesssim \exp(-L_{\boldsymbol{\ell}} n^{\beta/(\beta+d)} \Delta) + \exp(-3k_{1,n}/2)(1 + o(1)).$$

*Proof of Lemma 3.* In this proof, (i) we first evaluate the probability

$$\mathbb{P}_{\mathcal{D}_n}\left(\left|\frac{r_{v,n}}{r_{1,n}} - \ell_v\right| \geq \Delta\right) \tag{41}$$

Subsequently, (ii) evalute

$$\mathbb{P}_{\mathcal{D}_n}\left(\|\boldsymbol{z}_{n,\boldsymbol{k}}(X_*) - \boldsymbol{z}_{r\boldsymbol{\ell}}\|_\infty > \Delta\right) \tag{42}$$

by leveraging (41).

(i) For any positive sequence $\{b_n\}_{n\geq 1} \subset \mathbb{R}_{>0}$, we define $k'_{v,n} := \min\{k \in [n] \mid \|X_{(k)} - X_*\|_2 \geq \ell_v b_n\}$. Although the corresponding radius $r'_{v,n} := \|X_{(k'_{v,n})} - X_*\|_2$ is computed through the sequence $\{b_n\}$, it coincides with $r_{v,n} := \|X_{(k_{v,n})} - X_*\|_2$ as $b_n = r_{1,n}$ will be specified later.

For any $v \in \{2, 3, \ldots, V\}$, it holds that

$$\begin{aligned}
\mathbb{P}_{\mathcal{D}_n}\left(\left|\frac{r'_{v,n}}{b_n} - \ell_v\right| \geq \Delta\right) &= \mathbb{P}_{\mathcal{D}_n}\left(r'_{v,n} - b_n\ell_v \geq b_n\Delta\right) \\
&= \mathbb{P}_{\mathcal{D}_n}\left(r'_{v,n} \geq b_n(\ell_v + \Delta)\right) \\
&= \mathbb{P}_{\mathcal{D}_n}(\forall i \in [n], X_i \notin B(X_*; r'_{v,n}) \setminus B(X_*; b_n\ell_v)) \\
&\leq \mathbb{P}_{\mathcal{D}_n}(\forall i \in [n], X_i \notin B(X_*; b_n(\ell_v + \Delta)) \setminus B(X_*; b_n\ell_v)).
\end{aligned} \tag{43}$$

Considering a random variable $Z_i := \mathbb{1}(X_i \notin B(X_*; b_n(\ell_v + \varepsilon)) \setminus B(X_*; b_n\ell_v))$, that i.i.d. follows a Bernoulli distribution whose expectation is

$$\begin{aligned}
q_n &= 1 - \int_{B(X_*;b_n(\ell_v+\varepsilon))\setminus B(X_*;b_n\ell_v)} \mu(X)\mathrm{d}X \\
&\leq 1 - \mu_{\min} \int_{B(X_*;b_n(\ell_v+\Delta))\setminus B(X_*;b_n\ell_v)} \mathrm{d}X \\
&\leq 1 - \mu_{\min} \frac{\pi^{d/2}}{\Gamma(d/2+1)} b_n^d \{(\ell_v + \Delta)^d - \ell_v^d\} \\
&\leq 1 - \underbrace{\mu_{\min} \frac{\pi^{d/2}}{\Gamma(d/2+1)} d\ell_v^{d-1}}_{=:L_v} b_n^d \Delta,
\end{aligned}$$

(43) can be evaluated as

$$(43) = \mathbb{P}(Z_i = 1, \forall i \in [n]) = \mathbb{P}(Z_i = 1)^n = q_n^n \leq (1 - L_v b_n^d \Delta)^n. \tag{44}$$

By leveraging (44) and specifying $b_n = r_{1,n}$, we hereinafter evaluate (41). For any sequence $\{a_n\}_{n\geq 1} \subset \mathbb{R}_{>0}$,

$$\begin{aligned}
\mathbb{P}_{\mathcal{D}_n}\left(\left|\frac{r_{v,n}}{r_{1,n}} - \ell_v\right| \geq \Delta\right) &= \int_0^\infty \mathbb{P}_{\mathcal{D}_n}\left(\left|\frac{r'_{v,n}}{b_n} - \ell_v\right| \geq \Delta\right) \mathbb{P}_{\mathcal{D}_n}(r_{1,n} = b_n)\, \mathrm{d}b_n \\
&\leq \left\{\int_0^{a_n} + \int_{a_n}^\infty\right\} \mathbb{P}_{\mathcal{D}_n}\left(\left|\frac{r'_{v,n}}{b_n} - \ell_v\right| \geq \Delta\right) \mathbb{P}_{\mathcal{D}_n}(r_{1,n} = b_n)\, \mathrm{d}b_n \\
&\leq \underbrace{\mathbb{P}_{\mathcal{D}_n}\left(\left|\frac{r_{v,n}}{b_n} - \ell_v\right| \geq \Delta \mid b_n > a_n\right) \mathbb{P}_{\mathcal{D}_n}(r_{1,n} > a_n)}_{\leq(1-L_1 a_n^d \Delta)^n} \\
&\qquad + \mathbb{P}_{\mathcal{D}_n}\left(\left|\frac{r_{v,n}}{b_n} - \ell_v\right| \geq \Delta \mid b_n \leq a_n\right) \mathbb{P}_{\mathcal{D}_n}(r_{1,n} \leq a_n) \\
&\lesssim \underbrace{(1 - L_v a_n^d \Delta)^n}_{(\star1)} + \underbrace{\mathbb{P}_{\mathcal{D}_n}(r_{1,n} \leq a_n)}_{(\star2)}.
\end{aligned}$$

By specifying $a_n := n^{-1/(\beta+d)}$, the terms $(\star1), (\star2)$ are evaluated as follows.

(a) Regarding $(\star1)$, it holds that

$$(\star1) = (1 - L_1 n^{-d/(\beta+d)}\Delta)^n \leq \exp\left(-n^{\beta/(\beta+d)} L_1 \Delta\right),$$

as $(1 - 1/a)^b \leq ((1 - 1/a)^a)^{b/a} \leq \exp(-1)^{b/a} = \exp(-b/a)$ for all $a, b > 0$.

(b) Here we evaluate the second term ($\star 2$): considering a random variable $Z_i := \mathbb{1}(X_i \in B(X_*; a_n))$ that i.i.d. follows a Bernoulli distribution whose expectation is

$$q_n' := \int_{B(X_*;a_n)} \mu(X)\mathrm{d}X \le \mu_{\min} \int_{B(X_*;a_n)} \mathrm{d}X \le \mu_{\min} \frac{\pi^{d/2}}{\Gamma(d/2+1)} a_n^d \lesssim n^{-d/(\beta+d)},$$

we have an inequality

$$\mathbb{P}(r_{1,n} \le a_n) = \mathbb{P}\left(\sum_{i=1}^n Z_i \ge k_{1,n}\right) = \mathbb{P}\left(\sum_{i=1}^n Z_i \ge nq_n' + \lambda\right) \quad \text{(where } \lambda := k_{1,n} - nq_n')$$

$$\le \exp\left(-\frac{\lambda^2}{2(nq_n + \lambda/3)}\right)$$

$$\le \exp\left(-\frac{(k_{1,n} - nq_n')^2}{2(nq_n' + (k_{1,n} - nq_n')/3)}\right)$$

$$\lesssim \exp(-3k_{1,n}/2)(1 + o(1)) \qquad (\because nq_n' = o(k_{1,n}))$$

by referring to a Chernoff bound (Chung & Lu, 2006, Theorem 2.4) with $\mathbb{E}_{\mathcal{D}_n}(\sum_{i=1}^n Z_i) = nq_n'$.

Therefore, above (a) and (b) yield

$$\mathbb{P}_{\mathcal{D}_n}\left(\left|\frac{r_{v,n}}{r_{1,n}} - \ell_v\right| \ge \Delta\right) \lesssim \exp\left(-n^{\beta/(\beta+d)} L_1 \Delta\right) + \exp\left(-3k_{1,n}/2\right)(1 + o(1)). \tag{45}$$

(ii) We second evaluate (42). As it holds that

$$(\mathbf{1}^\top(\boldsymbol{I} - \mathcal{P}_{\boldsymbol{R}_{n,k}})\mathbf{1})(\mathbf{1}^\top(\boldsymbol{I} - \mathcal{P}_{\boldsymbol{R}})\mathbf{1})\|\boldsymbol{z}_{n,k} - \boldsymbol{z}_{r_{1,n}\ell}\|$$

$$= (\mathbf{1}^\top(\boldsymbol{I} - \mathcal{P}_{\boldsymbol{R}_{n,k}})\mathbf{1})(\mathbf{1}^\top(\boldsymbol{I} - \mathcal{P}_{\boldsymbol{R}})\mathbf{1})\left\|\frac{\boldsymbol{I} - \mathcal{P}_{\boldsymbol{R}_{n,k}}}{\mathbf{1}^\top(\boldsymbol{I} - \mathcal{P}_{\boldsymbol{R}_{n,k}})\mathbf{1}} - \frac{\boldsymbol{I} - \mathcal{P}_{\boldsymbol{R}}}{\mathbf{1}^\top(\boldsymbol{I} - \mathcal{P}_{\boldsymbol{R}})\mathbf{1}}\right\|_\infty$$

$$= \|(\mathbf{1}^\top(\boldsymbol{I} - \mathcal{P}_{\boldsymbol{R}_{n,k}})\mathbf{1})(\boldsymbol{I} - \mathcal{P}_{\boldsymbol{R}}) - (\mathbf{1}^\top(\boldsymbol{I} - \mathcal{P}_{\boldsymbol{R}})\mathbf{1})(\boldsymbol{I} - \mathcal{P}_{\boldsymbol{R}_{n,k}})\|_\infty$$

$$\le \|(\mathbf{1}^\top(\boldsymbol{I} - \mathcal{P}_{\boldsymbol{R}_{n,k}})\mathbf{1})(\boldsymbol{I} - \mathcal{P}_{\boldsymbol{R}}) - (\mathbf{1}^\top(\boldsymbol{I} - \mathcal{P}_{\boldsymbol{R}})\mathbf{1})(\boldsymbol{I} - \mathcal{P}_{\boldsymbol{R}})\|_\infty$$

$$\qquad\qquad + \|(\mathbf{1}^\top(\boldsymbol{I} - \mathcal{P}_{\boldsymbol{R}})\mathbf{1})(\boldsymbol{I} - \mathcal{P}_{\boldsymbol{R}}) - (\mathbf{1}^\top(\boldsymbol{I} - \mathcal{P}_{\boldsymbol{R}})\mathbf{1})(\boldsymbol{I} - \mathcal{P}_{\boldsymbol{R}_{n,k}})\|_\infty$$

$$\le |\mathbf{1}^\top(\mathcal{P}_{\boldsymbol{R}_{n,k}} - \mathcal{P}_{\boldsymbol{R}})\mathbf{1}|\|(\boldsymbol{I} - \mathcal{P}_{\boldsymbol{R}})\|_\infty + |\mathbf{1}^\top(\boldsymbol{I} - \mathcal{P}_{\boldsymbol{R}})\mathbf{1}|\|\mathcal{P}_{\boldsymbol{R}_{n,k}} - \mathcal{P}_{\boldsymbol{R}}\|_\infty.$$

$$\le \|\mathbf{1}\|_\infty^2 \|\boldsymbol{I} - \mathcal{P}_{\boldsymbol{R}}\|_\infty \|\mathcal{P}_{\boldsymbol{R}} - \mathcal{P}_{\boldsymbol{R}_{n,k}}\|_\infty,$$

there exist constants $L^{(1)}, L^{(2)} > 0$ such that

$$\|\boldsymbol{z}_{n,k}(X_*) - \boldsymbol{z}_{r\ell}\| \le L^{(1)}\|\mathcal{P}_{\boldsymbol{R}_n} - \mathcal{P}_{\boldsymbol{R}}\|_\infty \le L^{(2)}\|\boldsymbol{r}_n/r_{1,n} - \boldsymbol{\ell}\|_\infty,$$

where $\boldsymbol{r}_n = (r_{1,n}, r_{2,n}, \ldots, r_{V,n}) \in \mathbb{R}^V$.

Consequently, above (i) and (ii) yield

$$\mathbb{P}\left(\|\boldsymbol{z}_{n,k}(X_*) - \boldsymbol{z}_{r\ell}\|_\infty > \Delta\right) \le \mathbb{P}(L^{(2)}\|\boldsymbol{r}_n/r_{1,n}\|_\infty > \Delta)$$

$$\lesssim \exp(-L_\ell n^{\beta/(\beta+d)}\Delta) + \exp(-3k_n/2)(1 + o(1)).$$

for some constant $L_\ell > 0$. $\qquad\qquad\qquad\qquad\qquad\qquad\qquad\qquad\qquad\qquad\qquad\qquad\qquad\Box$

---

**Lemma 4** (Evaluation for (34)). Let

- $X_* \in \mathcal{S}(\mu), \beta > 0, t \in [0, 1], i_o \in \mathbb{N}$,

- $L_\beta^{***} := L_\beta^{**} \tilde{L}^{-\beta/d}$, where $\tilde{L} := (\sup_{X \in \mathcal{X}} \mu(X)) \frac{\pi^{d/2}}{\Gamma(d/2+1)}$ and $L_\beta^{**}$ is defined in Lemma 2.

- $\Delta_o := L_\beta^{***} t^{\beta/d}, \Delta_{i_o} := 2^{i_o} \Delta_o.$

If $\Delta(X_*) > \Delta_{i_o}$, it holds that $X_* \notin \partial_{t, \Delta(X_*) - \Delta_{i_o}}$.

---

*Proof of Lemma 4.* For any $r \in (0, \tilde{r}_t(X_*)]$,

$$t \leq \int_{B(X_*;r)} \mu(X)\mathrm{d}X \leq \left(\sup_{X \in \mathcal{X}} \mu(X)\right) \int_{B(X_*;r)} \mathrm{d}X = \left(\sup_{X \in \mathcal{X}} \mu(X)\right) \frac{\pi^{d/2}}{\Gamma(d/2+1)} r^d = \tilde{L} r^d. \qquad (46)$$

Assuming that $\eta(X_*) > 1/2$ without loss of generality, we have

$$\begin{aligned}
\rho_{r\boldsymbol{\ell}}^{(\infty)}(X_*) &\geq \eta(X_*) - L_\beta^{**} r^\beta && (\because \text{Lemma 2}) \\
&\geq \eta(X_*) - L_\beta^{**}(\tilde{L}^{-1/d}t^{1/d})^\beta && (\because \text{ineq. (46)}) \\
&= \eta(X_*) - (L_\beta^{**}\tilde{L}^{-\beta/d})t^{\beta/d} \\
&= \eta(X_*) - \Delta_o && (\because \Delta_o = (L_\beta^{**}\tilde{L}^{-\beta/d})t^{\beta/d}) \\
&= \eta(X_*) - 2^{-i_o}\Delta_{i_o} && (\because \Delta_{i_o} = 2^{i_o}\Delta_o) \\
&= \frac{1}{2} + (\Delta(X_*) - 2^{-i_o}\Delta_{i_o}) && (\because \Delta(X_*) = |\eta(X_*) - 1/2|, \eta(X_*) > 1/2) \\
&\geq \frac{1}{2} + (\Delta(X_*) - \Delta_{i_0}) && (\because \Delta_{i_o} \geq 2^{-i_o}\Delta_{i_o})
\end{aligned}$$

for any $r \in (0, \tilde{r}_t(X_*)]$; it means that $X_* \in \mathcal{X}_{t,\Delta(X_*)-\Delta_{i_o}}^+$, whereupon $X_* \notin \partial_{t,\Delta(X_*)-\Delta_{i_o}}$. Similar holds for the case $\eta(X_*) < 1/2$. Thus we have proved $X_* \notin \partial_{t,\Delta(X_*)-\Delta_{i_o}}$. $\qquad\square$

---

**Lemma 5** (Evaluation for (35)). Let $X_* \in \mathcal{X}, \Delta \in [0,1/2]$ and $r_{1,n} := \|X_{(k_{1,n})} - X_*\|_2$. Then, it holds for $C_1 = 1/8V^2L_{\boldsymbol{z}}^2$ that

$$\mathbb{P}_{\mathcal{D}_n}\left(|\rho_{n,\boldsymbol{k}}(X_*) - \rho_{r_{1,n}\boldsymbol{\ell}}^{(\infty)}(X_*)| \geq \Delta/2\right)$$
$$\lesssim \exp(-C_1 k_{1,n}\Delta^2) + \exp(-L_{\boldsymbol{\ell}} n^{\beta/(\beta+d)}\Delta) + \exp(-3k_{1,n}/2)(1+o(1)).$$

---

*Proof of Lemma 5.* By simply decomposing the terms, we have

$$|\rho_{n,\boldsymbol{k}}(X_*) - \rho_{r_{1,n}\boldsymbol{\ell}}^{(\infty)}(X_*)| = |\underbrace{\boldsymbol{z}_{n,\boldsymbol{k}}(X_*)^\top \boldsymbol{\varphi}_{n,\boldsymbol{k}}(X_*)}_{=\rho_{n,\boldsymbol{k}}(X_*)} - \boldsymbol{z}_{r_{1,n}\boldsymbol{\ell}}^\top \boldsymbol{\varphi}_{n,\boldsymbol{k}}(X_*)| \qquad (47)$$

$$+ |\boldsymbol{z}_{r_{1,n}\boldsymbol{\ell}}^\top \boldsymbol{\varphi}_{n,\boldsymbol{k}}(X_*) - \boldsymbol{z}_{r_{1,n}\boldsymbol{\ell}}^\top \boldsymbol{\varphi}_{r_{1,n}\boldsymbol{\ell}}^{(\infty)}(X_*)| \qquad (48)$$

where the terms (47), (48) are evaluated as follows.

(i) Regarding the first term (47),

$$(47) = |\{\boldsymbol{z}_{n,\boldsymbol{k}}(X_*) - \boldsymbol{z}_{r_{1,n}\boldsymbol{\ell}}\}\boldsymbol{\varphi}_{n,\boldsymbol{k}}(X_*)| \leq \|\boldsymbol{z}_{n,\boldsymbol{k}}(X_*) - \boldsymbol{z}_{r_{1,n}\boldsymbol{\ell}}\|_\infty \underbrace{\|\boldsymbol{\varphi}_{n,\boldsymbol{k}}(X_*)\|_\infty}_{\leq 1}.$$

Therefore, Lemma 3 leads to

$$\begin{aligned}
\mathbb{P}((47) \geq \Delta/4) &\leq \mathbb{P}(\|\boldsymbol{z}_{n,\boldsymbol{k}}(X_*) - \boldsymbol{z}_{r_{1,n}\boldsymbol{\ell}}\|_\infty \geq \Delta/4) \\
&\lesssim \exp(-L_{\boldsymbol{\ell}} n^{\beta/(\beta+d)}\Delta) + \exp(-3k_n/2)(1+o(1)),
\end{aligned}$$

for some constant $L_{\boldsymbol{\ell}} > 0$.

(ii) Regarding the second term (48),

$$(48) = |\boldsymbol{z}_{r\boldsymbol{\ell}}^\top\{\boldsymbol{\varphi}_{n,\boldsymbol{k}}(X_*) - \boldsymbol{\varphi}_{r\boldsymbol{\ell}}^{(\infty)}(X_*)\}| \leq \underbrace{\|\boldsymbol{z}_{r\boldsymbol{\ell}}\|_\infty}_{\leq L_{\boldsymbol{z}}} \sum_{v=1}^V |\varphi_{n,k_v}(X_*) - \varphi_{rh_v(X_*)}^{(\infty)}(X_*)|,$$

and Chaudhuri & Dasgupta (2014) Lemma 9 proves that

$$\mathbb{P}\left(|\varphi_{n,k_v}(X_*) - \varphi_{r_v}^{(\infty)}(X_*)| \geq \Delta/4VL_{\boldsymbol{z}}\right) \lesssim \exp(-2k_v(\Delta/4VL_{\boldsymbol{z}})^2).$$

Therefore, we have

$$\mathbb{P}((48) \geq \Delta/4) \lesssim \mathbb{P}\left(L_{\boldsymbol{z}} \sum_{v=1}^V |\varphi_{n,k_v}(X_*) - \varphi_{r_v}^{(\infty)}(X_*)| \geq \Delta/4\right)$$

$$\leq \sum_{v=1}^{V} \mathbb{P}\left(|\varphi_{n,k_v}(X_*) - \varphi_{r_v}^{(\infty)}(X_*)| \geq \Delta/4VL_{\boldsymbol{z}}\right)$$

$$\lesssim \exp(-2k_1\Delta^2/(4VL_{\boldsymbol{z}})^2) = \exp(-k_1 C_1 \Delta^2),$$

with $C_1 := 1/8V^2 L_{\boldsymbol{z}}^2$.

Considering above evaluations, we have

$$\mathbb{P}(|\rho_{n,\boldsymbol{k}}(X_*) - \eta(X_*)| \geq \Delta/2) \leq \mathbb{P}((47) \geq \Delta/4) + \mathbb{P}((48) \geq \Delta/4)$$

$$\lesssim \exp(-C_1 k_1 \Delta^2) + \exp(-L_{\boldsymbol{\ell}} n^{\beta/(\beta+d)}\Delta) + \exp(-3k_n/2)(1+o(1)).$$

The assertion is proved. $\qquad\square$

---

**Lemma 6** (Evaluation for (36)). Let $X_* \in \mathcal{X}$ and $\Delta \in [0, 1/2]$. Then, it holds for $C_2 = 1/(2VL_{\boldsymbol{z}})^2 (= C_1/2)$ that
$$\mathbb{P}_{\mathcal{D}_n}\left(|\rho_{n,\boldsymbol{k}}(X_*) - \eta(X_*)| \geq \Delta/2\right) \lesssim \exp\left(-C_2 k_{1,n}\Delta^2\right) + \exp(-n).$$

---

*Proof of Lemma 6.* By simply decomposing the terms, we have

$$|\rho_{n,\boldsymbol{k}}(X_*) - \eta(X_*)| \leq \underbrace{|\rho_{n,\boldsymbol{k}}(X_*) - \rho_{r_{1,n}\boldsymbol{\ell}}^{(\infty)}(X_*)|}_{(\star 1)} + \underbrace{|\rho_{r_{1,n}\boldsymbol{\ell}}^{(\infty)}(X_*) - \eta(X_*)|}_{(\star 2)}. \tag{49}$$

- Regarding the first term $(\star 1)$, applying Lemma 5 immediately leads to

$$\mathbb{P}((\star 1) \geq \Delta/4) \lesssim \exp(-C_2 k_{1,n}\Delta^2),$$

  where $C_2 := C_1/2 = 1/16V^2 L_{\boldsymbol{z}}^2$.

- Here, we consider the second term $(\star 2)$. As Lemma 2 shows that $|\rho_{r\boldsymbol{\ell}}^{(\infty)}(X_*) - \eta(X_*)| \leq L_{\beta}^{**} r_{1,n}^{\beta}$, we have

$$\mathbb{P}(|\rho_{r_{1,n}\boldsymbol{\ell}}^{(\infty)}(X_*) - \eta(X_*)| \geq \Delta/2) \leq \mathbb{P}(L_{\beta}^{**} r_{1,n}^{\beta} \geq \Delta/2) = \mathbb{P}(r_{1,n} \geq (\Delta/2L_{\beta}^{**})^{1/\beta}) \tag{50}$$

  (50) represents the probability that less than $k_{1,n}$ out of $n$ feature vectors lie in a region $B(X_*; \Delta_*)$ with $\Delta_* := (\Delta/2L_{\beta}^{**})^{1/\beta}$; considering a random variable $Z_i := \mathbf{1}(X_i \in B(X_*; \Delta_*))$, that i.i.d. follows a Bernoulli distribution whose expectation is $q_* := \int_{B(X_*;\Delta_*)} \mu(X)\mathrm{d}X > 0$,

$$(50) = \mathbb{P}\left(\bar{Z}_n < \frac{k_{1,n}}{n}\right) \leq \mathbb{P}\left(|\bar{Z}_n - q_*| \geq q_* - \frac{k_{1,n}}{n}\right) \leq 2\exp\left(-2n\left(q_* - \frac{k_{1,n}}{n}\right)^2\right)$$

  by Höeffding's inequality. As $\frac{k_{1,n}}{n} \asymp n^{-d/(2\beta+d)} \leq q_*/2$ for sufficiently large $n$, we have $(50) \lesssim \exp(-n)$.

Considering above $(\star 1)$ and $(\star 2)$

$$\mathbb{P}(|\rho_{n,\boldsymbol{k}}(X_*) - \eta(X_*)| \geq \Delta) \leq \mathbb{P}((\star 1) \geq \Delta/2) + \mathbb{P}((\star 2) \geq \Delta/2) \lesssim \exp(-C_2 k_{1,n}\Delta^2) + \exp(-n)$$

for some $C_2 > 0$; the assertion is then proved. $\qquad\square$

---

**Lemma 7** (Evaluation for (37)). Let $X_* \in \mathcal{X}, t, \delta \in [0, 1]$ and $k \in [(1-\delta)nt]$. Then,
$$\mathbb{P}_{\mathcal{D}_n}(\|X_{(k)} - X_*\|_2 > \tilde{r}_t(X_*)) \lesssim \exp(-k\delta^2/2).$$

---

*Proof of Lemma 7.* The assertion is obtained by Chaudhuri & Dasgupta (2014) Lemma 8. $\qquad\square$