[Reviews · NeurIPS 2020]

Review 1

Summary and Contributions: This paper proposes a computationally tractable way of computing weights (possibly negative) for weighted k-NN classification that achieves the same rate of convergence as the optimal weights approach of Samworth (2012). The proposed method first computes unweighted k-NN estimators for different values of k, and then solves a regression problem that can interestingly be interpreted as extrapolating to the imaginary k=0 nearest neighbor.

Strengths: The proposed multiscale k-NN approach is interesting and, I suspect, would mainly be of interest to ML researchers working on theory for nonparametric methods. The impact is a bit limited since it really is focused on weighted k-NN classification where weights can be negative. The work is very heavily based on Samworth's 2012 optimal k-NN weights paper. The proposed method seems to do about as well as one of Samworth's methods.

Weaknesses: In terms of hyperparameter tuning, in your experimental results, you specify hyperparameters values for V, c_1, c_2, c_3, ... c_V (in a way to match Theorem 2's conditions). Is there some general strategy for how to tune these hyperparameters including how to set V? Understanding how sensitive multiscale k-NN is to how these are tuned would be very helpful to understand. Overall, the proposed multiscale k-NN approach is quite clever although it seems to only barely do better than Samworth's baseline with nonnegative weights (see Table 2; where multiscale k-NN does better than Samworth's baseline with nonnegative weights, the improvement is generally very small).

Correctness: The claims and method appear correct.

Clarity: The paper is decently written.

Relation to Prior Work: Yes.

Reproducibility: Yes

Additional Feedback: Update after reading author feedback and other reviews: the author feedback sufficiently addressed my concerns, and I am increasing my score to a 6. Please revise the paper according to what was in the author response to improve the paper draft.


Review 2

Summary and Contributions: The paper considers binary classification using weighted k-nearest neighbours. The main research problem of interest has to do with convergence rates for the excess risk (in expectation over the sample) achieved by different implementations of weighted k-NN. As a key background work, they cite work by Samworth (2012) which shows that by using real-valued weights instead of being constrained to non-negative weights, it is possible, at least in principle, to achieve a faster rate which is essentially optimal among all classifiers. The problem is that actual computation of weights which satisfy the conditions put forward by Samworth is a challenge. With this basic context in place, the authors propose a new algorithmic approach doing real-valued weighted k-NN which is easy to implement, and their main claim is that this practical alternative achieves the same rate as the Samworth-type weight setting. They also provide empirical results comparing existing k-NN weighting schemes with two representative implementations of their proposed procedure on a number of benchmark datasets, for fixed values of sample size n.

Strengths: I would say that the main strength of this paper is that the proposed algorithm (multi-scale weighted k-NN) is easy to understand, straightforward to implement, and that the technical material presented in this paper is for the most part very easy to follow. The procedure is rather intuitive as well, and so as a contribution of a new general-purpose method for doing weighted k-NN with a theoretical guarantee tacked on, it may be appealing to practitioners.

Weaknesses: I would say that the chief weakness of this work is that the novel results presented here seem rather limited in terms of new insights. The vast majority of the paper is dedicated to background review; there is about 1.5 pages of new technical material here, and a half-page of empirical analysis which does not consider changing values of n (i.e., no actual comparison of rates). The proof of their main result may be of value from a technical standpoint, but the authors themselves state that it follows very closely from existing results (Chaudhury and Dasgupta, 2014), and all details are relegated to the supplement.

Correctness: The formal result appears to be correct, though I have not looked in detail at the proof given in supplementary materials. The empirical tests conducted here are relevant to their general problem of interest (classification using weighted k-NN), although direct comparison of learning curves (performance over n) might have been nice to see, given that the main focus of the formal side of the paper is a method that achieves near-optimal rates.

Clarity: The paper is written clearly.

Relation to Prior Work: The paper gives a detailed background on related work, and it is clear where the present work stands in the existing literature.

Reproducibility: Yes

Additional Feedback: - As an additional comment, I thought that most of the paper was pleasant to read for a wide audience, save for lines 271-276, which happen to be the key technical conditions used in Theorem 2, the main theoretical result of this paper. I think it would benefit the paper greatly to provide some more discussion with these conditions, to make the level of clarity on par with the rest of the paper. - One small point: in 2.2 (notation), line 120, there is a typo (bold x instead of X). After reading the feedback and the other reviews, I maintain my current assessment.


Review 3

Summary and Contributions: The paper deals with the problem of achieving improvement over convergence rate of “vanilla” k-NN. This is previously done by leveraging assumptions over the distribution (its smoothness (\beta-Holder condition, \gamma-neighbour average smoothness) and “margin” (\alpha-margin condition, namely, upper bounding the probability of instances whose expected label expectation is ~1/2)) along with using weighted k-NN, with several methods for defining the weights and their resulting convergence rate. The paper proposes a new method, called MS-k-NN, of estimating several unweighted k-NN estimators, for different values of k (\nu(k) for simplicity of notation in the review). For each k, a radius r is associated by taking the distance from the query to its k-closest neighbour. Then, pairs (r(k), \nu(k)) are obtained, and parameters b are obtained by linear regression in order to estimate \nu(k) by a polynomial in r(k). It is proven that this method obtains the optimal convergence rates obtained by more cumbersome methods of weighted k-NN. It is shown that experimentally, over a few datasets, the performance of MS-k-NN is similar to that of the weighted k-NN.

Strengths: The method proposed in the paper is intuitive and very simple. It also provides interesting insight as to what lies beneath the optimal weights in weighted k-NN methods by showing correspondence between the weights obtained in MS-k-NN and in those methods.

Weaknesses: The method seems to provide little benefit over LP estimators. Although, as mentioned in the paper, the number of weights in the regression is reduced by a factor of 2, to 0.5\beta, it is not clear whether this is significant in real world problems, where large value of \beta seems to be an unrealistic assumption. Additionally, they’re also very easy to implement.

Correctness: seems correct.

Clarity: clear.

Relation to Prior Work: yes

Reproducibility: Yes

Additional Feedback:

[Author Response · NeurIPS 2020]

We greatly appreciate your efforts and precious time for providing us with helpful comments and fascinating ideas!

**R1: Contribution is limited as this work is heavily based on Samworth (2012). Furthermore, Samworth (2012) suggests using cross-validation (CV); we may find good choices numerically without explicit formulas.**

As written in line 92, we first note that the weights obtained via multiscale $k$-NN are DIFFERENT from Samworth (2012) as shown in Figure 2 in Supplement F, though they attain the same convergence rate (in terms of the order w.r.t. $n$). Although multiscale $k$-NN is heavily compared with Samworth (2012) as it is only one baseline in the same setting, these two methods are based on different ideas: Taylor-series of the risk is directly optimized in Samworth (2012) whereas multiscale $k$-NN considers a regression. Secondly, Samworth (2012) chooses only $k$ by CV (with the weights $\boldsymbol{w}_k = (w_1, w_2, \ldots, w_k)$, that can be explicitly obtained only for the limited cases $\beta \in \{2, 4\}$). Namely, equations should be solved to obtain the weights $\boldsymbol{w}_k$; the issue of Samworth (2012) remains, even if CV is utilized. Although we may conduct CV to choose $\boldsymbol{w}_k$ directly from $\mathcal{W}^k$ (for some set $\mathcal{W} \subset \mathbb{R}$ with $|\mathcal{W}| = m$), it requires the computational complexity $O(m^k)$, which is too large to compute in practice (as $k \geq 10^2$ in many practical cases). Therefore, multiscale $k$-NN still has advantages, compared to Samworth (2012) equipped with CV. As these are confusing points which are not well explained, we would like to revise the current manuscript to describe these advantages clearer.

**R1: Is there some general strategy to set the parameters including $V$?**

The multiscale $k$-NN attains the improved convergence rate for **any** combination of $V > \lfloor \beta/2 \rfloor + 1$ and $\ell_1 < \ell_2 < \cdots < \ell_V$, as the weights $\boldsymbol{w}_k$ are automatically adapted to the setting of $(V, \ell_1, \ell_2, \ldots, \ell_V)$ via the regression. However, the smoothness $\beta$ of the underlying function $\eta$ cannot be obtained in practice; this is a common problem with the local polynomial (LP) regression. From both theoretical and application perspectives, we may simply employ a large $V$ (e.g., $V = 100$) so that $V$ is expected to be larger than $\lfloor \beta/2 \rfloor + 1$. Even if such a large $V$ is employed, the computational complexity for the regression remains very small, as the number of regression coefficients to be estimated is only $1 + V$. It is different from the LP regression, as LP leverages $1 + d + d^2 + \cdots + d^V$ terms to attain the same convergence rate.

**R1, R2: There is only a limited empirical analysis.**

Although the main purpose of this paper is to provide an intuitive idea to understand how to obtain a faster convergence rate, we agree with this comment; we will add some experiments to emphasize the advantages of the improved rate.

**R2: Whereas the proof may be of value of this paper, the authors state that it follows closely from Chaudhuri and Dasgupta (2014), and all details are relegated to the supplement.**

The most important point is how to evaluate the asymptotic bias (and variance); our Theorem 1 indicates the order of the reduced bias of the multiscale $k$-NN, though its straightforward proof is mostly based on tedious Taylor expansion, which may not be deserving of explanation in detail in the main body. This proof is independent of Chaudhuri and Dasgupta (2014). Once the asymptotic bias (and variance) are obtained, following Chaudhuri and Dasgupta (2014) almost yields Theorem 2 (as well as many of nonparametric theories), though there remain some tedious technical issues to be considered. We will revise the current proof sketch so that the proof can be grasped more easily.

**R2: The vast majority of the paper is dedicated to background review.**

Although the detailed background review is necessary for explaining the position of this paper (among papers written with different notations/assumptions), we would like to revise the manuscript to condense the descriptions.

**R2: It would benefit the paper greatly to provide some more discussion with the conditions in Theorem 2.**

We agree with your comment. We would like to add some explanations (discussions) on the conditions. For instance, for (C-2): it is for selecting $k_1, k_2, ..., k_V$ so that the corresponding $r_1, r_2, ..., r_V$ distribute with regular intervals.

**R3: The method seems to provide little benefit over LP estimators.**

Whereas LP estimator considers the polynomial function $g(x)$ defined for $x \in \mathbb{R}^d$, multiscale $k$-NN considers $f(r)$ defined for the radius $r > 0$; the numbers of coefficients to be estimated are $1 + d + d^2 + \cdots + d^V$ for LP and $1 + V$ for multiscale $k$-NN for attaining the same convergence rate. Furthermore, multiscale $k$-NN is expected to inherit the favorable properties of the $k$-NN, such as the adaptability to the underlying manifold of the data vectors (Cheng and Wu, 2013). We are thinking about proving such favorable properties in future works.

# References

Chaudhuri, K. and Dasgupta, S. (2014). Rates of convergence for nearest neighbor classification. In *Advances in Neural Information Processing Systems 27*, pages 3437–3445. Curran Associates, Inc.

Cheng, M.-y. and Wu, H.-t. (2013). Local linear regression on manifolds and its geometric interpretation. *JASA*, 108(504):1421–1434.

Samworth, R. J. (2012). Optimal weighted nearest neighbour classifiers. *Ann. Statist.*, 40(5):2733–2763.


[Meta-Review · NeurIPS 2020]

The paper presents a new nonparametric learning method, which seems to combine certain elements of k-nearest neighbors with elements of local regression estimation. It recovers the optimal rates for classification with smooth regression functions and Tsybakov noise, previously established for a local polynomial regression method, but uses a predictor representation involving far fewer parameters, as in a simple weighted k-NN predictor. The reviewers favor accepting the paper. However, they have some reservations, as they would prefer the paper be presented differently, with more space dedicated to presenting the new techniques, and with more investigation into the strengths of this particular method compared to the well-known standard techniques.